# ACRBP (Sp32) is involved in priming sperm for the acrosome reaction and the binding of sperm to the zona pellucida in a porcine model

Yoku Kato [‡¤a], Satheesh Kumar[‡¤b], Christian Lessard, Janice L. Bailey[¤c]*

Département des sciences animales, Centre de recherche en reproduction, développement et santé intergénérationnelle, Université Laval, Québec, Canada

¤a Current address: Vitrolife KK, Shibakōen, Minato-ku, Japan
¤b Current address: Department of Veterinary Biochemistry, Madras Veterinary College, Tamilnadu Veterinary and Animal Sciences University, Madras, India
¤c Current address: Fonds de recherche du Québec en Nature et technologies, Québec, Canada
‡ These authors are co-first authors as they equally contributed to this work.
* janice.bailey@frq.gouv.qc.ca

**Data Availability Statement:** All relevant data are within the manuscript and its Supporting Information files.

## Abstract

In boar sperm, we have previously shown that capacitation is associated with the appearance of the p32 tyrosine phosphoprotein complex. The principal tyrosine phosphoprotein involved in this complex is the acrosin-binding protein (ACRBP), which regulates the auto-conversion of proacrosin to intermediate forms of acrosin in both boar and mouse sperm. However, the complete biological role of ACRBP has not yet been elucidated. In this study, we tested the hypothesis that tyrosine phophorylation and the presence of the ACRBP in the sperm head are largely necessary to induce capacitation, the acrosome reaction (AR) and sperm-zona pellucida (ZP) binding, all of which are necessary steps for fertilization. *In vitro* fertilization (IVF) was performed using matured porcine oocytes and pre-capacitated boar sperm cultured with anti-phosphotyrosine antibodies or antibodies against ACRBP. Anti-ACRBP antibodies reduced capacitation and spontaneous AR ($P<0.05$). Sperm-ZP binding declined in the presence of anti-phosphotyrosine or anti-ACRBP antibodies. The localisation of anti-ACRBP antibodies on the sperm head, reduced the ability of the sperm to undergo the AR in response to solubilized ZP or by inhibiting the sarco/endoplasmic reticulum $Ca^{2+}$-ATPase. These results support our hypothesis that tyrosine phosphorylated proteins and ACRBP are present upon the sperm surface in order to participate in sperm-ZP binding, and that ACRBP upon the surface of the sperm head facilitates capacitation and the AR in the porcine.

## Introduction

It has long been known that mammalian sperm undergo biochemical and physiological changes in the female reproductive tract [1, 2] or during incubation in appropriate medium *in*

**Funding:** This research was financed by the Natural Sciences and Engineering Research Council of Canada (NSERC) of Canada (Discovery Grant RGPIN-2017-06386 to JLB). YK was supported by institutional scholarships from the Faculté des sciences de l'agriculture et de l'alimentation and the Centre de recherche en reproduction, développement et santé intergénérationnelle.

**Competing interests:** The authors have declared that no competing interests exist.

*vitro* [3]. Following capacitation, sperm are able to bind to the egg zona pellucida (ZP) and undergo the AR which allows the sperm to penetrate the egg [4]. Capacitation is regulated by signal transduction pathways occurring during sperm transit through the female reproductive tract that involves protein phosphorylation at tyrosine residues in order to prime sperm for the AR [5].

Increased protein phosphorylation is associated with capacitation, hyperactivated motility, sperm-ZP binding, the AR, and sperm-oocyte binding and fusion [6]. In mice, humans, bovines, stallions and pigs, the capacitated state is correlated to increased protein tyrosine phosphorylation [7–13]. Particularly, the tyrosine phosphorylation of sperm head proteins has been suggested to facilitate protein translocation to ZP binding sites at the sperm head surface [14, 15]. Moreover, tyrosine phosphorylation on capacitated human sperm tails detected by immunofluorescence correlated strongly with sperm-ZP binding capacity but not with the ZP-induced AR [16]. In mammals, many proteins have been characterized as being differentially phosphorylated on tyrosine residues upon capacitation: A-kinase anchor protein 4 [17], A kinase-anchoring proteins 3 and 4 [18], dihydrolipoamide dehydrogenase [19], heat shock protein 90 [20], valosin containing protein [21], a calcium binding fibrous sheath protein [22], the mitochondrial phospholipid hydroperoxidase glutathione peroxidase [23], and pyruvate dehydrogenase E1 $\alpha$ [24]. In the pig model, our laboratory has demonstrated that capacitation is associated with the calcium-dependant appearance of a Mr 32 000 group of tyrosine phosphoproteins, named p32. The role and regulation of the p32 tyrosine phosphoprotein complex is unknown. It has been suggested that it is involved in the AR and/or sperm-oocyte binding since sperm surface tyrosine phosphorylation increases during capacitation and high $Ca^{2+}$ concentration enhanced tyrosine phosphorylation in the presence of p32. In pigs, it was confirmed that the 32 kDa proacrosin binding protein, ACRBP (also known as sp32 and OY-TES-1), is the principal tyrosine phosphorylated protein during capacitation [15]. Tandem mass spectrometry of the excised Mr 32 000 protein (p32) on a non-reducing/reducing gel revealed peptide homology with ACRBP.

ACRBP is expressed from primary spermatocytes to spermatozoa [25] and catalyses the conversion of proacrosin to acrosin, thereby enabling the AR [26]. ACRBP tyrosine phosphorylation is also seen in humans [18]. ACRBP also has a secondary sperm-ZP binding affinity domain [27], and is known to be a cancer-testis antigen [28–30]. In boar sperm, ACRBP is primarily produced in the form of a precursor of 58–60 kDa and cleaved into a mature form of 32 kDa [26, 31] that is phosphorylated during capacitation [15]. Transgenic mice, lacking protein convertase subtilisin/kexin type 4 displayed difficulties in converting the precursor form of ACRBP to mature ACRBP, which was accompanied by extensive male infertility [31]. A recent study highlighted the importance of ACRBP in human fertility, showing that an under-representation of ACRBP peptides in infertile men impaired capacitation [32]. In the porcine, ACRBP is assumed to be a component of head plasma membranes and is transported to the surface of acrosome-intact sperm during capacitation where it colocalizes with zonadhesin and proacrosin/acrosin [33]. In vitro fertilization (IVF) assays using cumulus-intact oocytes indicated that sperm produced by mice lacking the ACRBP gene have a reduced ability to fertilize oocytes. The fertilization rate for the deficient sperm was less than 10% of that of normal sperm and attributed to reduced capacity of the sperm to bind the ZP and fuse with the oolemma [34].

In the present study, we tested the hypothesis that tyrosine phosphoproteins and ACRBP are necessary for sperm capacitation, the AR, and fertilization in the porcine model. To this end, *in vitro* matured pig oocytes were inseminated with pre-capacitated fresh boar sperm in the presence of affinity-purified mouse anti-phosphotyrosine or rabbit anti-ACRBP antibodies to conduct sperm function assays. We demonstrate here that sperm surface tyrosine

phosphorylated proteins and ACRBP induce capacitation, priming the sperm for the AR through regulation of intracellular $Ca^{2+}$ through the sarco/endoplasmic reticulum $Ca^{2+}$-ATPase (SERCA) pump. Additionally, we show that tyrosine phosphoproteins and ACRBP are also involved in sperm-ZP binding.

## Material and methods

### Ethical statement

These studies were conducted with the approval of the institutional committees for research integrity, notably the Comité de protection des animaux de l'Université Laval and the Comité de gestion des produits chimiques de l'Université Laval.

### Chemicals

Monoclonal mouse anti-phosphotyrosine antibodies (clone 4G10), peroxidase-conjugated goat anti-mouse antibodies, and peroxidase-conjugated goat anti-rabbit antibodies were purchased from Millipore Corporate (Billerica, MA). Affinity-purified rabbit anti-ACRBP antibodies were a kind gift from Baba et al. and were prepared as described previously [26]. Tris (hydroxymethyl) aminoethane (Tris) was obtained from Bio-Rad (Mississauga, ON). Thapsigargin was obtained from Enzo Life Sciences (Ontario, Canada). Fluorescein isothiocyanate conjugated goat anti-rabbit antibody was sourced from Biosource International (Camarillo, Calif). Affinity purified polyclonal anti-SERCA 2 (N-19) goat antibody was obtained from Santa Cruz Biotechnology Inc. (Santa Cruz, CA). The secondary antibodies raised against goat and immunoglobulin G (IgGs) conjugated to fluorescein isothiocyanate were from Jackson ImmunoResearch Laboratories Inc. (West Grove, PA). Unless otherwise stated, all chemicals used in the present study were obtained from Sigma Chemical Company (St. Louis, MO).

### Media

The maturation medium for the *in vitro* maturation of pig oocyte was the BSA-free North Carolina State University (NCSU) 37 medium (110 mM NaCl, 5 mM KCl, 1 mM $KH_2PO_4$, 1mM L-glutamine, 5.5 mM D-glucose, 7 mM taurine, 5 mM hypotaurine, 1 mM $MgSO_4$, 1 mM $CaCl_2$, 25 mM $NaHCO_3$) supplemented with 25 mM beta-mercaptoethanol, 0.1 mg/ml cysteine, 10% (v:v) porcine ovarian follicular fluid (the same batch was used for all repetitions), 1 mM dibutyryl-cAMP and hormonal supplements (10 IU/ml human chorionic gonadotropin; hCG, Ayerst Laboratories Inc., Philadelphia, PA, and 10 IU/ml equine chorionic gonadotropin; eCG, Folligon; Intervet, Whitby, ON, Canada) as previously reported [35]. Porcine ovarian follicular fluid was collected from the follicles (4–6 mm in diameter) of porcine ovaries transported from a slaughterhouse [35]. The fertilization medium for *in vitro* fertilization was modified Tris-buffered medium (mTBM: 110 mM NaCl, 3 mM KCl, 20 mM Tris, 11 mM glucose, 5 mM sodium-pyruvate and 7.5 mM $CaCl_2$) containing 0.1% (w:v) BSA (Sigma, A-4503) and 100 mM caffeine [35]. The sperm washing medium before insemination [36] was the Beltsville Thawing Solution (BTS, 205 mM glucose, 20 mM sodium citrate, 15 mM $NaHCO_3$, 3.5 mM EDTA, 5 mM KCl, pH 7.0–7.2).

### Cumulus-oocyte complex preparation and *in vitro* maturation

Porcine ovaries were obtained from an abattoir, placed in saline, and transported to the laboratory within 1–3 hours. The collection and *in vitro* maturation of cumulus-oocyte complexes (COCs) were carried out as previously described [35–37]. Briefly, follicles of 4–6 mm in diameter were punctured and aspirated, then harvested oocytes with intact cumulus cells and evenly

granulated cytoplasm were selected (about 200–300 oocytes per repetition), and washed three times with 5 ml Hepes-buffered Tyrode medium (114 mM NaCl, 2 mM NaHCO$_3$, 3 mM KCl, 10 mM sodium lactate, 0.3 mM NaH$_2$PO$_4$, 0.5 mM MgCl$_2$, 2 mM CaCl$_2$, 12 mM sorbitol, 10 mM Hepes, 0.2 mM sodium pyruvate) containing 0.01% (w:v) polyvinyl alcohol in 60 mm petri dishes [38]. Around 50 selected COCs were transferred into 4-well dishes (Nunc, Roskilde, Denmark) containing 500 μl maturation medium covered with 200 μl mineral oil. The COCs were cultured for 20–22 hours at 38.6˚C under 5% CO$_2$ in the air with high humidity. Then, the COCs were washed with 5 ml maturation medium without dibutyrylc-AMP, hCG, and eCG in petri dishes three times and transferred into 4-well dishes (Nunc, Roskilde, Denmark) containing 500 μl of fresh maturation medium without dibutyrylc-AMP, hCG, and eCG, and covered with 200 ul mineral oil, and cultured for an additional 24 hours [39]. The proportion of *in vitro* oocyte maturation was greater than 90% in all the experiments.

## Sperm preparation and *in vitro* fertilization

Sperm washing and insemination were carried out as previously reported [38]. Briefly, semen samples were collected from different fertile boars at the Centre d'Insemination Porcine du Quebec (St Lambert, Canada) and transported to the laboratory, as formerly described [40]. The fresh semen was conserved in 10 ml of Gedil solution (GEDIL®, Genes Diffusion, Tournai, Douai, France) in a 50 ml plastic tube overnight. Conserved semen was layered over 2 ml of 45% Percoll solution (Amersham Biosciences Corp., Piscataway, NJ), centrifuged (5 minutes, 22˚C, 500 x G), and washed once with 10 ml BTS in a 15 ml centrifuge tube (10 minutes, 22˚C, 250 x G). Motility and concentration were evaluated using computer-aided sperm analysis (CASA), and the sperm pellet was then resuspended in fertilization medium and diluted with fertilization medium to a concentration of 500,000 motile sperm/ml in a 50 ml plastic tube. After completing *in vitro* maturation, all COCs were transferred into a 15 ml centrifuge tube containing 2 ml of maturation medium supplemented with 0.1% (w/v) hyaluronidase (Sigma Chemical Company, St Louis, Mo, H-3506) and cumulus cells were removed by vortexing. Oocytes were washed in 60 mm petri dishes with 3 ml fertilization medium three times and placed in 50 μl droplets of fertilization medium (25 oocytes/50 μl droplet) and covered with 200 ul mineral oil in 1-well of 4-well dishes. The dishes were incubated at 38.6˚C under 5% CO$_2$ in air with high humidity until the addition of the sperm. Aliquots of the sperm suspension (50 μl) were added to the 50 μl droplets of fertilization medium, giving a final sperm concentration of 250, 000 motile sperm/ml. Oocytes and sperm were coincubated for 6 hours at 38.6˚C under 5% CO$_2$ in the air with high humidity. Inseminated oocytes were then washed in 2 ml of fresh NCSU 37 supplemented with 1% BSA and without dibutyrylc-AMP, hCG, and eCG in 60 mm plastic culture dishes and transferred into each well of 4-well dishes containing 500 μl NCSU 37 supplemented with 1% BSA and without dibutyrylc-AMP, hCG, and eCG. The samples were covered with mineral oil and cultured for a further 12 hours at 38.6˚C under 5% CO$_2$ in air with high humidity.

## Evaluation of sperm capacitation, AR, and penetration

Sperm capacitation was determined by chlortetracycline staining as previously reported [41]. In brief, the sperm which had been cultured in fertilization medium were centrifuged at 600 g for 6 minutes. Fifteen μl of precipitated sperm were mixed thoroughly with 15 μl chlortetracycline solution containing chlortetracycline 750 μM, 5 mM cysteine, 130 mM NaCl, 20 mM Tris, and fixed with 0.2 μl 12.5% (w/v) paraformaldehyde in 0.5 M Tris-HCl (pH 7.4) in a 1 ml centrifuge tube. The mixture was then placed on a glass slide, covered with a coverslip and left in a moist environment in the dark at room temperature until observation. Two hundred

sperm were examined using a Nikon microscope equipped with fluorescent optics (excitation 400–440 nm: BV2-A filter). The chlortetracycline staining patterns were classified into three types: uncapacitated pattern, exhibiting bright fluorescence over the whole region of the sperm head; capacitated pattern, showing fluorescence on the sperm head but not over the post-acrosomal region; and acrosome-reacted pattern, exhibiting weak fluorescence observed over the sperm head with a bright band sometimes present in the equatorial region.

Acrosome-reacted sperm were evaluated using fluorescein-labeled *Pisum sativum* agglutinin (PSA-FITC) as previously described [41]. In brief, 15 μl of sperm which had been cultured in fertilization medium was smeared onto a slide, allowed to dry, then fixed and permeabilized in absolute ethanol for 10 minutes. After fixation, the smears were covered with 50 μl PSA-FITC (100 μg/ml) and placed in a dark, moist environment in the dark at room temperature for 30 minutes. The slides were rinsed with Tris-buffered saline (TBS; 213 mM NaCl, 50 mM Tris-HCl, pH 7.0), then mounted with glycerol and a coverslip. Two hundred sperm were counted and classified as "acrosome-intact" or "acrosome-reacted" with a Nikon microscope equipped with fluorescent optics (excitation 450–490 nm: B2-A filter).

The numbers of viable sperm were counted by eosin-nigrosin exclusion staining [41]. Fifteen μl of sperm suspension were mixed with 15 μl eosin-nigrosin solution in a 1 ml centrifuge tube and smeared onto a slide at room temperature. After drying, the samples were covered with permount and a coverslip. Two hundred sperm were evaluated under light microscope: live sperm showed no coloration and dead cells showed as pink coloration.

Fixation of the inseminated oocytes and evaluation of penetration were carried out as previously reported [35]. Briefly, to evaluate the pronuclear stage during fertilization, subsets of post-IVF oocytes were fixed on a slide with 3:1 acetic acid: ethanol and stained with 1% orcein. Oocytes considered to be fertilized were classified in three groups: normal fertilization (female pronucleus, male pronucleus and sperm tail); polyspermy (more than two pronucleus, two or more sperm tails in the cytoplasm with condensed heads, or two or more decondensed heads in the cytoplasm); and asynchrony, with a condensed sperm head (female pronucleus and a condensed sperm head). The experiments to investigate the effect of anti-phosphotyrosine and anti-ACRBP antibodies upon fertilization were repeated four times and five times, respectively.

## Sperm-ZP binding assay

Sperm-ZP binding assays were carried out as described previously [42]. Briefly, thirty minutes after insemination, unbound and loosely adhered spermatozoa were removed from oocytes using a fine pipette. Oocytes were then fixed in 2% glutaraldehyde and mounted on slides, and the numbers of sperm bound to the ZP were counted using a phase contrast microscope. The experiments to determine the effect of anti-phosphotyrosine and anti-ACRBP antibodies upon sperm-ZP binding were repeated three times.

## ZP collection and solubilization

Following *in vitro* maturation, all COCs were transferred into a 15 ml centrifuge tube containing 2 ml of maturation medium supplemented with 0.1% (w/v) hyaluronidase and cumulus cells were removed by vortexing. Oocytes were transferred and mixed with 10 μl acid TBS (pH 2.0) in 30 mm petri dishes to solubilize the ZP and 10 μl of TBS (pH 8.0) were then added to the oocytes to neutralize the pH. One hundred μl of solubilized ZP were collected from 500 oocytes, placed in a 500 μl centrifuge tube using a glass pipette, and stored in a– 20˚C freezer until required.

### Anti-phosphotyrosine and anti-ACRBP antibodies *in vitro* assays

Washed boar sperm were incubated in fertilization medium containing 1 μg/ml of anti-phosphotyrosine antibodies or 10 μg/ml of anti-ACRBP at 38.6˚C in an atmosphere of 5% $CO_2$ in air and 100% humidity. Pre-immune immunoglobulin and ortho-phosphotyrosine as a blocking peptide were used as negative controls and added to the fertilization medium to culture for 1 and 4 hours. After 1 and 4 hours of incubation with antibodies, capacitation and the AR were assessed by chlortetracycline and PSA-FITC staining. Sperm were smeared and dried on a glass slide and the number of viable sperm was determined by eosin/nigrosin exclusion staining. Then, *in vitro* matured porcine oocytes were inseminated with washed and uncultured sperm in fertilization medium for 6 hours. The number of sperm-ZP binding was observed by contrast microscopy after extensive washing to remove loosely bound sperm. The experiment to assess the effect of anti-phosphotyrosine antibodies upon capacitation was repeated four times and the experiment to determine the effect on the AR was replicated three times. Furthermore, the experiments to determine the effect of anti-ACRBP antibodies upon capacitation and the AR were repeated five times and six times, respectively.

### Induction of the AR with thapsigargin, calcium ionophore (A23187) and solubilized ZP

Fresh boar sperm were cultured in fertilization medium containing anti-ACRBP antibodies for 1 or 4 hours pre-incubation and the cultured sperm were supplemented with either only media, 0.1% DMSO (solvent control for A23187 and thapsigargin), 10 uM calcium ionophore (A23187; an AR inducer by the elevation of intracellular $Ca^{2+}$), 10 μM thapsigargin (non-competitive inhibitor of the SERCA pump) or 1 ZP/μl of solubilized ZP. The sperm were then cultured for an additional 15 minutes. Capacitation and the AR were assessed using PSA-FITC staining, and the numbers of viable sperm were determined by eosin-nigrosin exclusion staining. The experiment to determine on the impact of anti-ACRBP on the AR induced by thapsigargin and A23187 was replicated three times. In addition, the experiment to assess the effect of anti-ACRBP on the AR induction of boar sperm by solubilized ZP was replicated four times.

### Indirect immunofluorescence of ACRBP

Five hundred μl of washed boar sperm suspension ($8.0 \times 10^5$ spermatazoa/mL) were added to 0.5 ml of 4.0% formaldehyde in a 15 ml centrifuge tube and incubated for 1 hour. Following the fixation and wash, the sperm were then rinsed with 4 ml of PBS. The washed boar sperm suspension ($20 \times 10^6$ spermatozoa/mL) was allowed to settle for 30 min at room temperature on poly-L-lysine coated coverslips. Spermatozoa were permeabilized in 90% ethanol for 10 seconds and rinsed again with PBS. Non-specific sites were blocked with 10% non-immune goat serum prepared with PBS (blocking solution). Samples were incubated overnight at 4˚C with affinity-purified rabbit anti-ACRBP antibodies diluted in the blocking solution. They were then rinsed with PBS and incubated with fluorescein isothiocyanate conjugated goat anti-rabbit antibodies. For the control experiments, non-immune rabbit IgGs were used instead of the primary antibody. After washing with PBS, the coverslips were mounted on slides with 90% glycerol containing 1.5% DABCO as an anti-fading agent. Immunofluorescence was detected by epifluorescence microscopy with a UV light.

### Indirect immunofluorescence of SERCA

For indirect immunofluorescence of SERCA, sperm were cultured in capacitation medium. The medium used was a modified Krebs Ringer Bicarbonate medium (mKRB4.78 mM KCl,

1.19 mM KH$_2$PO4, 94.6 mM NaCl, 5.56 mM glucose, 25 mM NaHCO$_3$, 5 mM CaCl$_2$, 0.4% bovine serum albumin (BSA; type V, fatty acid-free), 0.5 mM pyruvate, and lactate 21.58 mM pH 7.4 [43].

The washed boar sperm suspension (20 x 10$^6$ spermatozoa/ mL) was allowed to settle for 30 min at room temperature on poly-L-lysine coated coverslips. Sperm were fixed for 15 min in 3.7% formaldehyde and then rinsed with PBS. Spermatozoa were permeabilized in 0.2% Triton X-100 for 10 min and rinsed again with PBS. Non-specific sites were blocked with 5% non-immune rabbit serum prepared with PBS (blocking solution). Samples were incubated over-night at 4˚C with the polyclonal SERCA 2 antibodies diluted in the blocking solution then rinsed with PBS and incubated with FITC-labelled rabbit anti-goat secondary antibody. For controls, non-immune goat IgGs were used instead of the primary antibody. After washing with PBS, coverslips were mounted on slides using 90% glycerol containing 1.5% DABCO as an anti-fading agent. Immunofluorescence was detected by epifluorescence microscopy with a UV light [44]

## Induction of the AR with thapsigargin, gingerol and BAPTA-K+

Sperm were cultured in capacitation medium (modified Krebs Ringer Bicarbonate as above) supplemented with or without 10 μM thapsigargin (non-competitive inhibitor of SERCA pumps), 50 μM gingerol (an activator of SERCA pumps), or 10mM BAPTA-K+ (a calcium chelator). Sperm aliquots were collected at 0, 20, 30, 60 and 180 minutes from the respective treatments. The AR was assessed using PSA-FITC staining. This experiment to explore how the SERCA inhibitor and activator induces the AR was repeated three times.

## Isolation of boar sperm proteins and Western immunoblotting

Proteins from boar sperm were extracted and isolated as described previously [40]. Sperm were washed with equal volume of sodium orthovanadate (0.2 mM final concentration) then centrifuged (3 minutes, 22˚C, 16060 x G). The resulting sperm pellets were resuspended in sample buffer without mercaptoethanol and heated for 1 min at 95˚C. The samples were then re-centrifuged (3 minutes, 22˚C, 16060 x G) and mercaptoethanol was added to a final concentration of 5% to the resulting supernatant, and the samples were stored immediately at -20˚C. Just prior to separation by SDS-PAGE, the sperm protein samples were heated for 1 min at 95˚C. Protein extraction with detergent Triton X-114 was carried out as reported previously [45]; the process involved a phase partitioning at 37˚C to separate the aqueous, integral membrane and insoluble matrix proteins.

The sperm proteins (1 x 10$^6$ washed spermatozoa/mL) were separated by 12% SDS–PAGE under reducing conditions [46] for western blotting with anti-ACRBP antibodies. The proteins were then transferred and non-specific binding sites on the membrane were blocked by incubating in TBST (20 mM Tris, pH 7.4, 0.9% (w/v) NaCl, and 0.1% (v/v) Tween-20) containing 5% (w/v) dry skim milk for 1 h at room temperature [47]. The membranes were incubated with anti-ACRBP antibody for 90 minutes at room temperature. After 3 X 10 min washes in TBST, the membranes were incubated in the presence of the corresponding HRP-conjugated secondary antibody for 1 hour at room temperature. The membranes were then washed, and the results were captured on x-ray films. Representative results from three repetitions are presented.

## Statistical analyses

All data from Tables 1 to 5 are expressed as the means ± SEM. The experiment data depicted in Fig 3 are expressed as the means ± SD. As specified above, all experiments were replicated

**Table 1. Effect of anti-phosphotyrosine and anti-ACRBP antibodies upon sperm-ZP binding[#].**

| Experimental Groups | Treatments | No. of ZP-bound sperm |
|---|---|---|
| Anti-phosphotyrosine antibodies | No antibody | 58.0 ± 5.6[a] |
| | Blocking peptide IgG | 52.8 ± 2.9[a] |
| | Anti-phosphotyrosine antibodies | 31.1 ± 2.0[b] |
| Anti-ACRBP antibodies | No antibody | 78.5 ± 9.1[c] |
| | Pre-immune rabbit IgG | 71.3 ± 10.6[c] |
| | Anti-ACRBP antibodies | 41.8 ± 4.1[d] |

[#] The experiments were repeated three times. Data represent the means ± SEM.

[*]Washed porcine sperm were cultured in fertilization medium with anti-phosphotyrosine antibodies or anti-ACRBP antibodies to evaluate sperm-ZP binding. Sperm were cultured with no antibodies, anti-phosphotyrosine antibodies bound ortho-phosphotyrosine as blocking peptide IgG (Malfunctioning anti-phosphotyrosine antibodies: as a negative control), or anti-phosphotyrosine antibodies. Sperm were treated with no antibodies, pre-immune rabbit IgG (negative control), or anti-ACRBP antibodies.

Different letters within vertical columns indicate significant differences as determined by one-way factorial analysis of variance with Fisher's protected least significant difference (a-b, c-d: P < 0.05).

three to five times. In all experiments, the capacitation, AR, and fertilization rates were evaluated for significance using one-way factorial analysis of variance with Fisher's protected least significant difference. Probability values greater than P = 0.05 were considered to be not statistically significant.

## Results

### Anti-phosphotyrosine and anti-ACRBP antibodies block sperm-ZP binding and fertilization

To assess the importance of sperm surface tyrosine phosphorylation and ACRBP on sperm-egg binding and fertility, *in vitro* matured pig oocytes were inseminated with fresh boar sperm in the presence of anti-phosphotyrosine or anti-ACRBP antibodies. The experiment was repeated three to five times. The treatment with anti-phosphotyrosine significantly reduced the numbers of incidences of sperm-ZP binding and the fertilization ratios (Tables 1 and 2,

**Table 2. Effect of anti-phosphotyrosine[#] and anti-ACRBP antibodies[$] upon fertilization[*].**

| Experimental Groups | Treatments | No. of inseminated oocytes | Normal Fertilization rate (%) | Polyspermy rate (%) |
|---|---|---|---|---|
| Anti-phosphotyrosine antibodies[*] | No antibody | 64 | 71.3 ± 5.0[a] | 2.3 ± 2.3 |
| | Blocking peptide IgG | 60 | 64.8 ± 6.6[a] | 7.9 ± 7.9 |
| | Anti-phosphotyrosine antibodies | 55 | 33.5 ± 6.2[b] | 3.1 ± 3.1 |
| Anti-ACRBP antibodies[**] | No antibody | 67 | 73.3 ± 7.7[c] | 7.2 ± 7.2 |
| | Pre-immune rabbit IgG | 63 | 61.1 ± 6.0[c] | 5.0 ± 5.0 |
| | Anti-ACRBP antibodies | 73 | 34.7 ± 3.6[d] | 2.5 ± 2.5 |

[#], [$] The experiments were repeated four and five times, respectively. Data represent the means ± SEM.

[*]Washed porcine sperm were cultured in fertilization medium with anti-phosphotyrosine antibodies or anti-ACRBP antibodies to evaluate fertilization.

[**]Sperm were cultured with no antibodies, anti-phosphotyrosine antibodies bound ortho-phosphotyrosine as blocking peptide IgG (Malfunctioning anti-phosphotyrosine antibodies: As a negative control), or anti-phosphotyrosine antibodies.

Sperm were treated with no antibodies, pre-immune rabbit IgG (negative control), or anti-ACRBP antibodies.

Different letters within vertical columns indicate significant differences as determined by one-way factorial analysis of variance with Fisher's protected least significant difference (a-b, c-d: P < 0.05).

P<0.05); and the treatment with anti-ACRBP antibodies impeded both of these processes (Tables 1 and 2, P<0.05).

## Anti-ACRBP antibodies impede capacitation and the AR as assessed by the chlortetracycline assay and PSA-FITC

Sperm were cultured in fertilization medium containing no antibodies, rabbit IgG (negative control), or anti-ACRBP antibodies for 1 and 4 hours to evaluate capacitation and the AR. The experiment with anti-phosphotyrosine antibodies was repeated four times to evaluate the effect on capacitation and three times for to assess the effect of these antibodies on the AR. In addition, the experiment with anti-ACRBP antibodies was repeated to five times to assess the effect on capacitation and six times for the AR. There was no decrease in the ratio of sperm exhibiting the particular pattern of capacitation by chlortetracycline staining when treated with anti-phosphotyrosine (Table 3, P>0.05). Moreover, a decrease in the proportion of sperm exhibiting the AR was observed (Table 3, P<0.05) between the group with no antibodies and the group with anti-phosphotyrosine, or blocking peptide IgG. However, the proportion of sperm presenting the characteristic capacitation pattern of chlortetracycline staining was lower when surface ACRBP were attached to specific ACRBP antibodies (Table 3, P<0.05). Furthermore, the proportion of sperm exhibiting AR decreased in the presence of anti-ACRBP antibodies (Table 3, P<0.05).

## Anti-ACRBP antibodies influence the AR in sperm cultured in fertilization medium supplemented with thapsigargin, as assessed by PSA-FITC

Sperm were cultured in fertilization media containing no antibodies, rabbit IgG (negative control), or anti-ACRBP antibodies for 1 and 4 hours and then supplemented with either only medium, dimethyl sulfoxide (DMSO: solvent control for A23187 and thapsigargin), A23187 (the AR inducer), or thapsigargin (the AR inducer) for each group respectively. After 15 minutes of culture with one of the above four chemical modulators, the mean AR was determined for each group (Table 4). The experiment was repeated three times. The average proportion of

**Table 3. Influence of anti-phosphotyrosine[#] and anti-ACRBP antibodies[$] upon sperm capacitation and the AR[$ $].**

| Experimental Groups | Treatments | Capacitation ratio (%) | AR ratio (%) |
|---|---|---|---|
| Anti-phosphotyrosine antibodies[*] | No antibody | 35.4 ± 4.3 | 45.5 ± 2.5 [a] |
| | Blocking peptide IgG | 35.3 ± 8.9 | 30.0 ± 2.2 [b] |
| | Anti-phosphotyrosine antibodies | 20.4 ± 4.6 | 32.6 ± 4.6 [b] |
| Anti-ACRBP antibodies[**] | No antibody | 43.3 ± 2.9[a] | 42.0 ± 2.5[a] |
| | Pre-immune rabbit IgG | 44.0 ± 4.7[a] | 38.2 ± 3.1[a] |
| | Anti-ACRBP antibodies | 23.8 ± 3.8[b] | 26.0 ± 4.4[b] |

Data represent the means ± SEM.

# The experiment was repeated four time for assessing capacitation and three times for evaluating the AR.

$ The experiment was repeated five time for assessing capacitation and six times for evaluating the AR.

[*]Washed porcine sperm were cultured in fertilization medium with anti-phosphotyrosine antibodies or anti-ACRBP antibodies, and stained with chlortetracycline staining and PSA-FITC to assess capacitation and the AR.

[**]Sperm were cultured with no antibodies, anti-phosphotyrosine antibodies bound ortho-phosphotyrosine as blocking peptide IgG (Malfunctioning anti-phosphotyrosine antibodies: As a negative control), or anti-phosphotyrosine antibodies. Sperm were cultured with no antibodies, pre-immune immunoglobulin (negative control), or anti-ACRBP antibodies.

$ $ Different letters within vertical columns indicate significant differences as determined by one-way factorial analysis of variance with Fisher's protected least significant difference (a-b: P < 0.05).

**Table 4. Impact of anti-ACRBP on the AR induced by thapsigargin and A23187[#].**

| Culture period | Treatments | Medium only | Dimethyl Sulfoxide | A23187 | Thapsigargin |
|---|---|---|---|---|---|
| 1 hour | No antibody | 37.8 ± 7.3 | 39.5 ± 4.5 | 77.5 ± 5.9 | 73.4 ± 4.5[a] |
| | Pre-immune rabbit IgG | 42.4 ± 8.0 | 37.8 ± 3.0 | 85.3 ± 4.3 | 73.6 ± 5.3[a] |
| | Anti-ACRBP antibodies | 32.0 ± 5.3 | 39.6 ± 5.8 | 77.7 ± 2.7 | 46.9 ± 3.4[b] |
| 4 hours | No antibody | 47.16 ± 6.4 | 45.2 ± 7.0 | 85.1 ± 4.3 | 72.4 ± 5.1[c] |
| | Pre-immune rabbit IgG | 46.2 ± 4.3 | 49.8 ± 14.1 | 75.6 ± 3.5 | 75.5 ± 1.8[c] |
| | Anti-ACRBP antibodies | 29.5 ± 29.5 | 42.2 ± 7.9 | 67.0 ± 4.2 | 38.8 ± 2.3[d] |

# The experiments were repeated three times. Data represent the means ± SEM.

*Washed porcine sperm were cultured in fertilization medium with no antibodies, pre-immune rabbit IgG (negative control), or anti-ACRBP rabbit antibodies for 1 and 4 hours. After the sperm were cultured, these sperm were treated with 1) no chemical modulators, 2) dimethyl sulfoxide (negative control; solvent control for A23187 and thapsigargin), 3) A23187 or 4) thapsigargin, and stained with the PSA-FITC to assess the AR (%).

Different letters within vertical columns indicate significant differences as determined by one-way factorial analysis of variance with Fisher's protected least significant difference (a-b, c-d: P < 0.05).

the characteristic AR-pattern of sperm in the culture supplemented with thapsigargin, after sperm were cultured with anti-ACRBP antibodies, was lower than the figure for sperm cultured with no antibodies (the negative control) or rabbit IgG for both 1 and 4 hours, respectively (Table 4, P<0.05). However, the AR rates for cultures supplemented with medium only, DMSO and A23187 were similar in the sperm cultures which contain only media, rabbit IgG, or anti-ACRBP antibodies after 1 and 4 hours (Table 4, P>0.05).

## Anti-ACRBP antibodies interfere with the AR in fertilization medium supplemented with thapsigargin and solubilized ZP, as assessed by PSA-FITC

Sperm were pre-cultured in the presence of no antibodies, rabbit IgG, or anti-ACRBP antibodies for 1 hour and then, medium only, DMSO, A23187, thapsigargin or solubilizedZP were added to the sperm culture medium for an additional 15 minutes to induce the AR (Table 5). The experiment was repeated four times. In the sperm culture with the addition of anti-ACRBP antibodies, the rate of the AR induced by thapsigargin was significantly lower (Table 5, P<0.05) than that of the cultures which contained no antibodies or pre-immune rabbit antibodies. In addition, the cultures with the addition of solubilizedZP displayed the same trend (Table 5, P<0.05). In contrast, the addition of 0.1% DMSO to the fertilization medium did not promote the parameters of the AR (Table 5, P>0.05). After 15 minutes in the

**Table 5. Effect of anti-ACRBP on the AR induction of boar sperm by solubilized ZP[#].**

| Treatments | Medium only | Dimethyl Sulfoxide | A23187 | Thapsigargin | Solubilized ZP |
|---|---|---|---|---|---|
| No antibody | 15.5 ± 2.1[a] | 16.7 ± 1.3 | 46.0 ± 4.3 | 55.1 ± 1.7[a] | 39.0 ± 2.3[a] |
| Pre-immune rabbit IgG | 14.9 ± 3.2[a] | 14.3 ± 1.4 | 43.1 ± 2.2 | 50.4 ± 0.8[a] | 38.7 ± 4.2[a] |
| Anti-ACRBP antibodies | 6.5 ± 1.6[b] | 13.9 ± 2.3 | 41.9 ± 4.0 | 32.7 ± 4.7[b] | 19..0 ± 4.0[b] |

# The experiments were repeated four times, Data represent the mean percentages ± SEM.

*Washed porcine sperm were cultured in fertilization medium with no antibodies, pre-immune rabbit IgG (negative control) or anti-ACRBP antibodies for 1 hour. After this, those sperm were cultured with the addition of 1) medium only, 2) dimethyl sulfoxide (negative control; solvent control for A23187 and thapsigargin), 3) A23187, 4) thapsigargin or 5) solubilized ZP, and stained with PSA-FITC to assess the AR (%).

Different letters within vertical columns indicate significant differences as determined by one-way factorial analysis of variance with Fisher's protected least significant difference (a-b, c-d: P < 0.05).

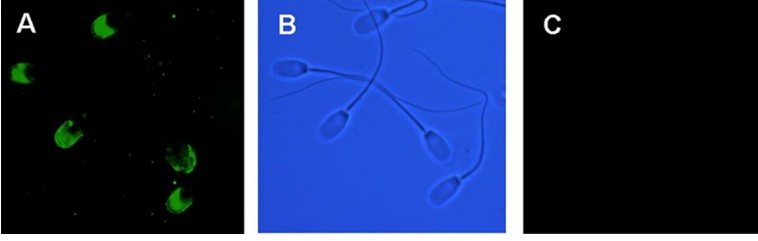

**Fig 1. Indirect immunolocalization of SERCA 2 in fixed and permeabilized porcine sperm.** Panel A indicates that acrosomal fluorescent labelling is evident when detected with SERCA 2 polyclonal antibody. Panel B shows the phase contrast microscopy photograph of the same field. Panel C shows the non-specific fluorescence of sperm labeled only with secondary fluorescein-conjugated rabbit anti-goat antibody. The experiment was repeated three times and representative photos are shown.

fertilization medium containing A23187, there was no difference in the AR ratios for sperm treated with no antibodies, rabbit antibodies, or anti-ACRBP antibodies (Table 5, P>0.05). There was a difference in the basal states of the AR between Tables 4 and 5, likely due to seasonal temperatures affects. The experiments carried out to produce Table 4 were performed during winter and the experiments performed to generate Table 5 were conducted during spring. Semen quality depends on the temperature of the environment where pigs are kept, and higher temperature negatively affects the semen quality [48].

## Localisation of SERCA 2 and ACRBP in boar sperm

Indirect immunofluorescence revealed the presence of SERCA 2 at the acrosome (Fig 1, panel A). No signal was observed in the post-acrosomal region or the mid-piece of the sperm. In control experiments, sperm incubated with the secondary antibodies only corroborate this observation (Fig 1, panel C). Indirect immunofluorescence showed the presence of ACRBP at the acrosome (Fig 2, panel A). ACRBP appears to be present on the head surface both before and after capacitation (Fig 2, panel A). Controls incubated only with the secondary antibodies confirmed these observations (Fig 2, panel C).

## Relationship between ACRBP and SERCA on the AR

The experiment to assess the effect of SERCA on the ACRBP shows that there is a corresponding decrease in ACRBP with thapsigargin treatment (Fig 3). To assess the effect of thapsigargin on the AR, the acrosomal status of the sperm treated with thapsigargin, gingerol, or BAPTA-K+ in capacitation medium was compared to that of the cultured sperm in capacitation medium



**Fig 2. Indirect immunolocalization of ACRBP in fixed and permeabilized porcine sperm.** Panel A indicates acrosomal fluorescent labelling detected with anti-ACRBP antibody. Panel B shows the phase contrast microscopy photo of the same field (Panel B). The non-specific fluorescence of sperm labeled with secondary fluorescein-conjugated rabbit anti-goat antibody only is displayed in Panel C. The experiment was repeated three times and representative photos are shown.

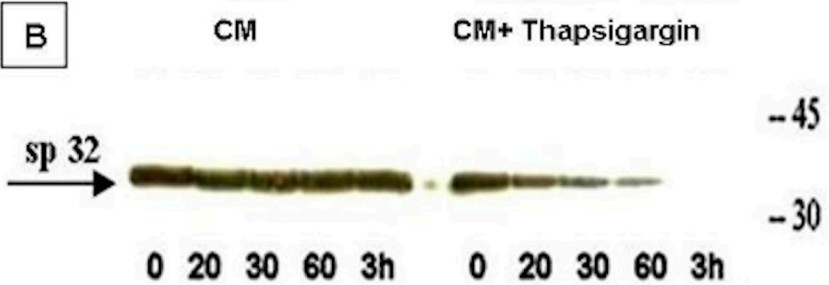

**Fig 3. Western blots of sperm proteins incubated in capacitation medium with or without thapsigargin probed with anti-ACRBP antibodies.** Washed boar sperm were incubated for 0, 20, 30, 60 and 180 minutes in capacitation medium with or without thapsigargin. The blot represents results from three samples.

without any chemical modulators (Fig 4). The percentage of sperm showing spontaneous AR in capacitation medium only was 6.66% (Fig 4). The presence of thapsigargin induced 29.16% of the sperm to undergo the AR. In contrast, the presence of gingerol and BAPTA-K+ caused around 10% and 1% of the sperm to undergo the AR, respectively. These data suggest that the induction of the AR is due to regulated calcium level increases in the thapsigargin-treated sperm from 20 minutes of incubation.

## Discussion

In support of the hypothesis, protein tyrosine phosphoprotein and ACRBP are important for capacitation, the AR, and fertilization in boar sperm. Indeed, sperm treated with

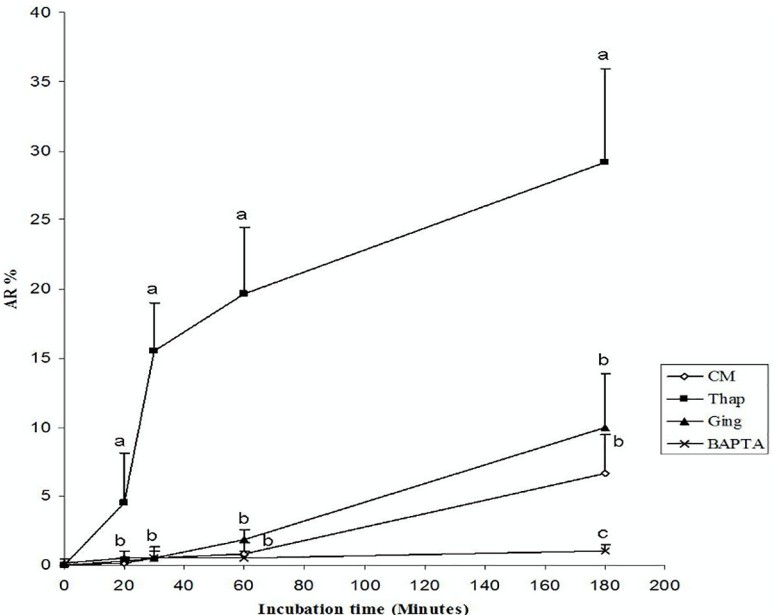

**Fig 4. Effect of sperm incubation in capacitation medium with thapsigargin, gingerol or BAPTA-K+ on the AR.** Washed porcine sperm were cultured in capacitation medium without any chemical modulators (control), or with thapsigargin, gingerol (Ging) or BAPTA-K+ (BAPTA) and stained with PSA-FITC to assess the AR (%). Different letters at 0, 20, 40, 60 and 180 minutes indicate significant differences as determined by one-way factorial analysis of variance with Fisher's protected least significant difference (a-d: $P < 0.05$). The experiments were replicated three times. Data represent the mean percentages ± SD.

anti-phosphotyrosine showed reduced sperm-ZP binding capacity and fertilization. The inclusion of anti-ACRBP antibodies to fertilization medium for the pre-culture of sperm before insemination also decreased capacitation, the AR, sperm-ZP binding capacity and fertilization. Interestingly, the presence of anti-ACRBP antibodies on the sperm surface prevented thapsigargin and solubilized ZP-induced AR. Together, these results suggest that surface sperm tyrosine phosphorylation and ACRBP are important for gamete interactions and that ACRBP influences $Ca^{2+}$ stores through the regulation of SERCA pumps and, ultimately, the regulation of calcium in the ZP induced-AR.

Our initial experiment demonstrates that anti-phosphotyrosine and anti-ACRBP antibodies significantly reduced sperm-ZP penetration (Table 1, $P < 0.05$) and fertilization (Table 2, $P < 0.05$). We had previously shown that ACRBP is one of a series of tyrosine-phosphorylated proteins associated with capacitation [15]. Serine/threonine and tyrosine phosphorylation are known to occur in spermatozoa, but only a few phosphorylated proteins, including hexokinase in the sperm head and the A-kinase-anchoring protein in the flagellum, have been formally identified [49]. Additionally, CatSper specifically controls protein tyrosine phosphorylation in intact mouse sperm in a time dependent manner [50] and two testis-specific aldolases were identified as substrates for tyrosine phosphorylation [51]. Additionally, Hsp60 and Eep99 on the surface membrane of the sperm forms molecular chaperone structures consisting of multiple functional receptors, thereby enhancing the sperm's capacity to bind to glycan molecules of the ZP [52]. The tyrosine phosphorylation of the sperm's surface is impaired by the attenuation of several signal transduction pathways, which in turn leads to decreased interaction between the sperm surface protein and ZP proteins [42]. This study shows that the extracellular signal-regulated kinase (ERK) module of the mitogen-activated protein kinase (MAPK) pathway induces the expression of surface phosphotyrosine in spermatozoa. Thr and Thr-Glu-Tyr protein motifs are present in the ACRBP amino acid sequences [26]. MAPK kinase phosphorylates Thr and Tyr residues within the Thr-Glus-Tyr motif, which is present in the active site of not only ERK 2 and ERK2, but also of ERK5 and ERKe7 [53]. Therefore, we suggest that these specific motifs for tyrosine phosphorylation are important for sperm-ZP biding.

Conversely, ACRBP and zonadhesin have a similar role in the initial sperm-ZP binding during capacitation [33]. Zonadhesin is expressed during spermatogenesis and is located at the nascent acrosome in the early spermatid [54, 55]. The function of this protein is to hypothetically aid membrane fusion at the onset of the AR and promote fusion to the ZP [56]. In fact, sperm-ZP binding was significantly decreased in the presence of zonadhesin antibodies during mouse capacitation [56]. Moreover, loss of zonadhesin in gene manipulated mice enables sperm to adhere to pig, cow, and rabbit ZP, and zonadhesin is indicated in species-specific ZP adhesion. Like zonadhesin, it is thought that ACRBP might contribute to the recognition of surface components on the ZP [31]. Indeed, ACRBP has been recognized as a secondary ZP-binding affinity domain [27]. Furthermore, ACRBP is expressed from primary spermatocytes to spermatozoa [25]. From these and our findings, we speculate that tyrosine phosphorylation and ACRBP on the sperm head surface might accelerate the binding of sperm to a specific structure within the ZP.

Anti-ACRBP antibodies obviously impeded capacitation (Table 3, $P < 0.05$). During capacitation, there was a clear elevation in $Ca^{2+}$ concentration [5] from several calcium channels, including the voltage-gated $Ca^{2+}$ channels, Catsper and transient receptor potential (TRP) [57, 58]. Notably, TRP channels regulate the major pathways involved in sperm capacitation [58], showing that capacitation enhances the translocation of TRP from the post-acrosomal region to the apical region of the sperm head. TRP channel activation in sperm, may involve store-operated $Ca^{2+}$ entry theory [59]. This theory suggests that the activation of phospholipase C is the trigger for the opening of TRP channels and $Ca^{2+}$ release from $Ca^{2+}$ stores [60]. TRP family

members were shown to be regulated by temperature, osmolarity, voltage, and pH [59]. From these findings, we speculate that ACRBP indirectly/directly activates calcium channels such as TRP through the SERCA pump to store calcium ions in the sperm. Therefore, we speculate that the anti-ACRBP antibodies affect calcium channels, showing that ACRBP influences capacitation and the AR.

Furthermore, anti-ACRBP antibodies impeded the AR (Table 3, P<0.05). While A23187 induced the AR in sperm treated with anti-ACRBP antibodies, anti-ACRBP antibodies interestingly prevented thapsigargin-induced AR (Table 4, P<0.05). ACRBP is transported to the surface of live acrosome-intact sperm during capacitation and interacts with other proteins on the sperm surface [33]. It was previously confirmed using immunofluorescence that ACRBP is present on the sperm head surface [15]. Thapsigargin is a non-competitive inhibitor of SERCA pumps [60] and is known to trigger the AR in mammalian sperm [61]. The presence of a SERCA-like pump in mammalian sperm, however, remains controversial [62] and we hereby report for the first time the presence of SERCA 2 in boar sperm acrosome using indirect immunofluorescence (Fig 1). SERCA 2 has previously been reported to be present on the acrosome and on the midpiece region of human, mouse and bovine sperm [44]. The presence of SERCA 2 at the acrosome suggests that this putative acrosomal store is important in the AR. The release of $Ca^{2+}$ from this intracellular store has been suggested to mediate opening of store operated channels [63] located at the plasma membrane, which causes a sustained cytosolic increase in the levels of $Ca^{2+}$ leading to the AR [60, 64]. Further, $IP_3$-receptors have been located in the acrosomal region in bovine, rat, hamster, mouse, dog, and human sperm [65–67]. It has been shown that the release of $Ca^{2+}$ from the acrosomal region during the AR is regulated through $IP_3$ receptors [68, 69]. In contrast, thapsigargin increases the intracellular $Ca^{2+}$ concentration in sperm and induces the AR, thereby resulting in $Ca^{2+}$ concentrations that are 10–300 times higher than those seen in somatic cells [70]. Low and high $Ca^{2+}$ levels are maintained in the cytosol and $Ca^{2+}$ stores respectively by SERCA pumps in sperm [71]. The addition of thapsigargin to human spermatozoa induced a dose-dependent increase in the percentage of AR in the presence of calcium [72], as well as progesterone- and thapsigargin-induced AR, which depend on an influx of calcium from the extracellular medium through store-operated calcium channels [73]. Therefore, we suggest that SERCA might be of central significance to the functioning of ACRBP via calcium regulation in sperm cells and the control of capacitation and the AR via calcium channels. Further experiments using confocal microscopy or proximity ligation assays would enhance our present findings.

Our present data indicate that ACRBP enhanced the solubilized ZP-induced AR (Table 5). Presumably, ACRBP interacts with $Ca^{2+}$ channels and ZP receptors on the sperm membrane surface. By immunostaining, we previously confirmed that anti-ACRBP antibodies bound to the sperm head surface [15] and that anti-ACRBP antibodies prevent sperm-ZP binding in the present study (Table 1). Sperm attachment to the ZP is also a trigger of the AR [61]. Current studies have shown that the ZP matrix of oocytes in several species is composed of 4 glycoproteins, namely, ZP1, ZP2, ZP3, and ZP4, in mice, pigs and cattle [74]. Interestingly, it was found that recombinant ZP3 enhanced the efflux of calcium from an intracellular $Ca^{2+}$ store during the beginning of the AR [75]. A model implicated $Ca^{2+}$ events via ZP3 signal transduction [59, 76], suggesting that ZP3 activation of a receptor induces the depolarization of the sperm membrane via low-voltage-activated $Ca^{2+}$ channels and activated phospholipase C, involving the activation of IP3-receptors on the AR by inositol trisphosphate. Following this, the absolute release of intracellular $Ca^{2+}$ stores produce a signal that activates TRP on the plasma membrane [59, 76]. In addition, co-immunoprecipitation indicated that proacrosin/acrosin, zonadhesin and ACRBP interact with each other and may travel from the acrosome area to the sperm surface [33]. Zonadhesin, which is present in the anterior head plasma membrane, was

demonstrated to interact with ZP3 suggesting that this protein has an initial role in sperm-ZP binding during capacitation [33]. Presumably, ACRBP interacts with these channels and ZP receptors on the sperm membrane surface.

It has been suggested that the inhibition of SERCA at the acrosome causes a rapid increase in the cytosolic calcium concentration [77]. The activation of SERCA by gingerol may have lowered the cytosolic calcium concentration, sequestering more $Ca^{2+}$ in the acrosome, thereby delaying the AR until 180 minutes (Fig 4). Furthermore, the presence of an extracellular chelator, BAPTA-K, completely blocked the AR. During the same period, the membrane when probed with anti-ACRBP antibodies showed a progressive decrease in the presence of ACRBP until 3h (Fig 3), which reflects this likely hydrolysis [26]. This indicates that ACRBP is related to the activity of SERCA in the beginning of the AR induction. These results establish that the increase in cytosolic calcium level is presumably caused by the inhibition of SERCA by thapsigargin [78]. The activation of SERCA by gingerol decreased the appearance of the calcium dependent AR after 180 min, which is related to the thapsigargin treatment (Fig 4). It was previously shown in human sperm that cytosolic calcium increase alone is not sufficient to trigger the AR, but that this process also requires the availability of intra-acrosomal $Ca^{2+}$ mediated through the IP3 receptors [26]. The release of acrosomal $Ca^{2+}$ has been shown to trigger protein tyrosine phosphorylation, leading to exocytosis [79].

The acrosomal status of the sperm was assessed by the PSA-FITC assay, which showed that the percentage of sperm which had undergone the AR was higher for the thapsigargin treated spermatozoa than that of cultured sperm in the capacitation medium without any chemicals, and the sperm cultured with gingerol or BAPTA-$K^+$ (Fig 4). The presence of gingerol in the capacitation medium caused a lower percentage of sperm to undergo the AR than with the addition of thapsigargin (Fig 4), this was likely due to SERCA activation by gingerol which would lower the cytosolic calcium concentration [79]. The chelation of extracellular calcium by BAPTA-$K^+$ could prevent the initial increase in internal calcium during capacitation [46], which simultaneously would impede the filling of the internal stores [79]. Even if some filling of the calcium stores occurs, chelation of extracellular calcium would also prevent the capacitative $Ca^{2+}$ influx through store-operated channels that would normally be activated after the emptying of internal $Ca^{2+}$ stores [79]. Extracellular calcium is thereby required for the AR. Recently, the presence of fusion pores which can connect the interior of the acrosome with the extracellular medium and the intra-acrosomal vesicles that are pinched off from the outer acrosomal membrane have been reported [80]. Future experiments are necessary to assess how ACRBP promotes the activation of the $Ca^{2+}$ channels with solubilized ZP, SERCA activators and inhibitors on the sperm surface using anti-ACRBP antibody.

In conclusion, this study has demonstrated that ACRBP and tyrosine phosphorylation are vital for sperm-ZP binding, and that ACRBP on the surface of the sperm head facilitates the AR through $Ca^{2+}$ regulation/channels also on the sperm head surface. This research is a step towards understanding the multiple functions of ACRBP and the mechanisms of ZP penetration in the porcine model. Future experiments are necessary to assess how ACRBP promotes the activation of the $Ca^{2+}$ channels on the sperm surface and the importance of ACRBP and its tyrosine phosphorylation in clinical trials on human spermatology and embryology.

## Supporting information

**S1 Table. The effect of anti-phosphotyrosine and anti-ACRBP antibodies upon sperm-ZP binding.**
(PDF)

**S2 Table. The effect of anti-phosphotyrosine# and anti-ACRBP antibodies upon fertilization.**
(PDF)

**S3 Table. The influence of anti-phosphotyrosine# and anti-ACRBP antibodies upon sperm capacitation and the acrosome reaction.**
(PDF)

**S4 Table. The impact of anti-ACRBP on the acrosome reaction induced by thapsigargin and A23187.**
(PDF)

**S5 Table. The effect of anti-ACRBP on the AR induction of boar sperm by soluble zona pellucida.**
(PDF)

**S6 Table. Effect of sperm incubation in capacitation medium with thapsigargin, gingerol or BAPTA-K+ on the AR (%).**
(PDF)

## Acknowledgments

We thank the Centre d'insemination porcine du Quebec for generously supplying boar semen, and Dr. Nicolas Santiquet, Isabelle Laflamme and Christine Guillemette for valuable technical advice.

## Author Contributions

**Conceptualization:** Yoku Kato, Satheesh Kumar, Janice L. Bailey.

**Data curation:** Yoku Kato, Satheesh Kumar, Christian Lessard, Janice L. Bailey.

**Formal analysis:** Yoku Kato, Satheesh Kumar, Janice L. Bailey.

**Funding acquisition:** Janice L. Bailey.

**Investigation:** Yoku Kato, Satheesh Kumar, Janice L. Bailey.

**Methodology:** Yoku Kato, Christian Lessard, Janice L. Bailey.

**Project administration:** Yoku Kato, Janice L. Bailey.

**Resources:** Yoku Kato, Christian Lessard, Janice L. Bailey.

**Software:** Janice L. Bailey.

**Supervision:** Janice L. Bailey.

**Validation:** Yoku Kato, Janice L. Bailey.

**Visualization:** Yoku Kato, Satheesh Kumar, Janice L. Bailey.

**Writing – original draft:** Yoku Kato, Satheesh Kumar, Janice L. Bailey.

**Writing – review & editing:** Yoku Kato, Janice L. Bailey.

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
