## [Decision Letter · Decision Letter 0]

15 May 2020

PONE-D-20-10428

ACRBP (Sp32) is involved in priming sperm for the acrosome reaction and the binding of sperm to the zona pellucida in a porcine model

Dear Dr. Y. Kato,

Thank you for submitting your manuscript to PLOS ONE. After careful consideration, we feel that it has merit but does not fully meet PLOS ONE’s publication criteria as it currently stands. Therefore, we invite you to submit a revised version of the manuscript that addresses the points raised during the review process.

To enhance the reproducibility of your results, we recommend that if applicable you deposit your laboratory protocols in protocols.io, where a protocol can be assigned its own identifier (DOI) such that it can be cited independently in the future. For instructions see: http://journals.plos.org/plosone/s/submission-guidelines#loc-laboratory-protocols

We look forward to receiving your revised manuscript.

Kind regards,

Joël R Drevet, Ph.D.

Academic Editor

PLOS ONE

Additional Editor Comments:

Please refer to the specific comments of each reviewer and try to modify your manuscript accordingly. Please note that at this stage, additional data is expected, as mentioned by the second reviewer, and not just cosmetic changes.

Journal Requirements:

We note that one or more of the authors are employed by a commercial company: Vitrolife KK.

3. Please amend the manuscript submission data (via Edit Submission) to include author Christian Lessard.

Reviewers' comments:

Reviewer's Responses to Questions

**Comments to the Author**

1. Is the manuscript technically sound, and do the data support the conclusions?

Reviewer #1: Yes

Reviewer #2: Yes

2. Has the statistical analysis been performed appropriately and rigorously? 

Reviewer #1: Yes

Reviewer #2: I Don't Know

3. Have the authors made all data underlying the findings in their manuscript fully available?

Reviewer #1: No

Reviewer #2: Yes

4. Is the manuscript presented in an intelligible fashion and written in standard English?

Reviewer #1: Yes

Reviewer #2: Yes

5. Review Comments to the Author

Reviewer #1: General comments:

The paper titled “ACRBP (Sp32) is involved in priming sperm for the acrosome reaction and the binding of sperm to the zona pellucida in a porcine model” has carried out experiments that are well-done methodologically, but it is difficult to understand the fundamental reasons that has led the authors to carry out the experiments. The justification for each experiment is not always present, despite being directly related to sperm capacitation. The number of replicates is not shown consistently, as it is absent sometimes or given an interval. There are many studies on sperm capacitation on porcine species, but these authors only refer to theirs or to other species, especially mice, where studies are carried out with sperm from the epididymis that have not been in contact with the seminal plasma and in many times these results are not comparable. Why have not the authors used more bibliographic references of pigs?

In summary, it is a work that could have been better written to make its understanding easier and relying more on authors who have worked on sperm capacitation in pigs.

Specific comments:

Line 185 washed once with ml BTS in a 15 ml centrifuge tube – time and centrifugation force?.

Line 203, should say number of oocytes and replicates used

Line 208, should explain what sperm suspension is

Line 273, should explain what is the pre-capacitated and pre-incubated boar sperm

Line 242, 250, 275, 288 should say the number of replicates,

Line 286 The authors wrote “Capacitation and the AR was assessed using PSA-FITC staining” but capacitation was evaluated by CTC. Specify.

The IVF results should be more extend and shown in a table, it is important to know how many oocytes were used, how many were penetrated, how many sperm penetrated per oocyte, polyspermy rate, asynchronous penetration... And, of course, the number of replicates included.

Reviewer #2: Referee’s comments:

The paper submitted by Kato et al. entitled “ACRBP (Sp32) is involved in priming sperm for the acrosome reaction and the binding of sperm to the zona pellucida in a porcine model” describes the role played by ACRBP on ZP-binding and acrosome reaction in relations with tyrosine phosphorylation and intracellular calcium concentration.

The experiments are clearly described, the results sustain the working hypothesis and the paper is globally well written. I only have a few remarks the authors may consider before the paper can be considered for publication:

MAJOR COMMENTS:

1-/ The fact that antibodies linked on the sperm surface inhibit sperm zona-pellucida binding is not a guarantee that the targeted protein is a key player in the fertilization process, as was proven in past for different candidate (such as the Crisp proteins). The most interesting action o ACRBP shown in this paper may thus be the potential interaction with SERCA. In this regards, even though the authors mention in their conclusion that “Future experiments are necessary to assess how ACRBP promotes the activation of the Ca2+ channels on the sperm surface”, it would strengthen the current work to show the co-localisation of these two proteins during the capacitation process. Maybe the authors could perform simple co-localization IF studies using confocal microscopy or proximity ligation assays? This would not impeach a more mechanistic study as proposed in the conclusion (with co-IP and intracellular calcium measurements maybe?).

2-/ There are differences in the basal states of acrosome reaction between figures 3 and 4. In the “medium only” condition, the ratio is around 40% in figure 3A whereas it barely reaches 20% in figure 4A. In figure 4A, there is a significant effect when anti-ACRBP antibodies are added, which is not commented in the “results” section. There are also huge differences in the percentages of AR when A23187 is added between these two figures (50% less in figure 4C versus 3C). Maybe the conditions are different and I did not get a point, unless, could the authors clarify the reasons of these discrepancies?

3-/ In figure 2A, the difference between “No antibodies” and “Blocking peptide” is around 50% (40% vs. 20%, respectively) with small SEM as indicated by the error bars. There is no significant difference indicated here, this seems curious… Could the authors verify this point?

4-/ Line 430-431, the authors write that “only a few phosphorylated proteins have been formally identified” concerning sperm tyrosine and/or Ser/Thr phosphorylated proteins. Even if the amount of literature is not huge, two papers in mouse spermatozoa report the identification of tyrosine phosphorylated proteins during capacitation (Arcelay et al., 2008 and Chung et al., 2014). Other data are available in different conditions and in different organisms, including Human. If this point needs to be discussed, some more recent and accurate references should be added and commented.

MINOR COMMENTS:

- Line 282 the authors indicate the use of 5mM A23187, is that correct? This concentration seems very high and A23187 is more usually used in the micromolar range.

- Lines 394-395 the authors write that “the additions of 0.1% DMSO in the fertilization medium promoted the parameters of the AR (Fig 4 B, P>0.05)”. Is this a mistake? Nothing seems different from the control situatin in this figure.

Conclusion: Considering the above-mentioned remarks, I propose major revisions before this article can be considered for publication.

6. PLOS authors have the option to publish the peer review history of their article (what does this mean?). If published, this will include your full peer review and any attached files.

Reviewer #1: No

Reviewer #2: No

---

## [Author Response · Author response to Decision Letter 0]

7 Nov 2020

[Comments and Questions] 

1. Editor Comment: Please refer to the specific comments of each reviewer and try to modify your manuscript accordingly. Please note that at this stage, additional data is expected, as mentioned by the second reviewer, and not just cosmetic changes. Journal Requirements: When submitting your revision, we need you to address these additional requirements. Please ensure that your manuscript meets PLOS ONE's style requirements, including those for file naming. The PLOS ONE style templates can be found at 

https://journals.plos.org/plosone/s/file?id=wjVg/PLOSOne_formatting_sample_main_body.pdf
https://journals.plos.org/plosone/s/file?id=ba62/PLOSOne_formatting_sample_title_authors_affiliations.pdf

Reply: Changed all the font of head to italic font. Please note L20, L64, L121, L123, L140, L158, L180, L209, L250, L259, L269, L286, L302, L318, L344, L352, L354, L397, L436, L474, L516, L540, L666, L672, and L677. 

2. Editor Comment: Thank you for stating the following in the Competing Interests section. "The authors have declared that no competing interests exist." We note that one or more of the authors are employed by a commercial company: Vitrolife KK. 

Reply: This research was financed by NSERC of Canada. They did not play a role in the study design, data collection and analysis, decision to publish, or preparation of the manuscript at all. Centre de recherche en reproduction, développement et santé intergénérationnelle and Faculté des sciences de l'agriculture et de l'alimentation have provide the salary of YK from 31th March, 2010 to 31th May 2011. These organizations did not play a role in the study design, data collection and analysis, decision to publish, or preparation of the manuscript and did only provided financial support in the form of authors' salaries and/or research materials. Currently, I am being employed by Vitrolife KK. However, they did not involve anything in this stud at all. When this study was conducted, I did not work for Vitrolife KK which provide me the salary now. 

3. Editor Comment: Please also include the following statement within your amended Funding Statement. “The funder provided support in the form of salaries for authors [insert relevant initials], but did not have any additional role in the study design, data collection and analysis, decision to publish, or preparation of the manuscript. The specific roles of these authors are articulated in the ‘author contributions’ section.” If your commercial affiliation did play a role in your study, please state and explain this role within your updated Funding Statement.

Reply: This research was financed by NSERC of Canada. They did not play a role in the study design, data collection and analysis, decision to publish, or preparation of the manuscript at all. Centre de recherche en reproduction, développement et santé intergénérationnelle and Faculté des sciences de l'agriculture et de l'alimentation have provide the salary of YK. These organizations did not play a role in the study design, data collection and analysis, decision to publish, or preparation of the manuscript and did only provided financial support in the form of authors' salaries and/or research materials. Currently, I have been employed by Vitrolife KK. However, they did not involve anything in this stud at all. When this study was conducted, I did not work for Vitrolife KK which provide the salary now. 

4. Editor Comment: Please also provide an updated Competing Interests Statement declaring this commercial affiliation along with any other relevant declarations relating to employment, consultancy, patents, products in development, or marketed products, etc. Within your Competing Interests Statement, please confirm that this commercial affiliation does not alter your adherence to all PLOS ONE policies on sharing data and materials by including the following statement: "This does not alter our adherence to PLOS ONE policies on sharing data and materials.” (as detailed online in our guide for authors http://journals.plos.org/plosone/s/competing-interests). If this adherence statement is not accurate and there are restrictions on sharing of data and/or materials, please state these. Please note that we cannot proceed with consideration of your article until this information has been declared. 

Reply: There are no competing interests statement declaring this commercial affiliation along with any other relevant declarations relating to employment, consultancy, patents, products in development, or marketed products at all. 

5. Editor Comment: Please include both an updated Funding Statement and Competing Interests Statement in your cover letter. We will change the online submission form on your behalf. Please know it is PLOS ONE policy for corresponding authors to declare, on behalf of all authors, all potential competing interests for the purposes of transparency. PLOS defines a competing interest as anything that interferes with, or could reasonably be perceived as interfering with, the full and objective presentation, peer review, editorial decision-making, or publication of research or non-research articles submitted to one of the journals. Competing interests can be financial or non-financial, professional, or personal. Competing interests can arise in relationship to an organization or another person. Please follow this link to our website for more details on competing interests: http://journals.plos.org/plosone/s/competing-interests

Reply: Our bullet letter includes both an updated Funding Statement and Competing Interests Statement. This research was financed by NSERC of Canada. They did not play a role in the study design, data collection and analysis, decision to publish, or preparation of the manuscript at all. Centre de recherche en reproduction, développement et santé intergénérationnelle and Faculté des sciences de l'agriculture et de l'alimentation have provide the salary of YK. These organizations did not play a role in the study design, data collection and analysis, decision to publish, or preparation of the manuscript and did only provided financial support in the form of authors' salaries and/or research materials. Currently, I am being employed by Vitrolife KK. However, they did not involve anything in this stud at all. When this study was conducted, I did not work for Vitrolife KK which provide the salary now. Furthermore, we all did have no competing interests statement declaring this commercial affiliation along with any other relevant declarations relating to employment, consultancy, patents, products in development, or marketed products at all.

6. Editor Comment: Please amend the manuscript submission data (via Edit Submission) to include author Christian Lessard.

Reply: I added Satheesh Kumar and Christian Lessard in the submission data. Please note the submission data.

7. Reviewers' comment: Is the manuscript technically sound, and do the data support the conclusions? The manuscript must describe a technically sound piece of scientific research with data that supports the conclusions. Experiments must have been conducted rigorously, with appropriate controls, replication, and sample sizes. The conclusions must be drawn appropriately based on the data presented. 

Reviewer #1: Yes 

Reviewer #2: Yes 

Reply: Thank you very much for your checking our paper.

8. Reviewers' comment: Has the statistical analysis been performed appropriately and rigorously?

Reviewer #1: Yes

Reviewer #2: I Don't Know

9. Reply: I changed all the figures to the tables to clearly show all the value and the number of experiment replicates and statistical values in the tables. I hope that these tables are strong enough to state that the statistical analysis been performed appropriately and rigorously. Please note all the tables.

10. Reviewers' comment: Have the authors made all data underlying the findings in their manuscript fully available? The PLOS Data policy requires authors to make all data underlying the findings described in their manuscript fully available without restriction, with rare exception (please refer to the Data Availability Statement in the manuscript PDF file). The data should be provided as part of the manuscript or its supporting information, or deposited to a public repository. For example, in addition to summary statistics, the data points behind means, medians and variance measures should be available. If there are restrictions on publicly sharing data—e.g. participant privacy or use of data from a third party—those must be specified. 

Reviewer #1: No

Reviewer #2: Yes

Reply: I changed all figures to tables. This indicates that all the value and the number of experiment replicates and statistical values are shown in the tables. I hope that these clear tables are strong enough to convince Reviewer #1. Please note the tables, L366-L367, L382-L383, L422-L423, L460-L461, L499-500.

11. Reviewers' comment: Is the manuscript presented in an intelligible fashion and written in standard English? PLOS ONE does not copyedit accepted manuscripts, so the language in submitted articles must be clear, correct, and unambiguous. Any typographical or grammatical errors should be corrected at revision, so please note any specific errors here. 

Reviewer #1: Yes

Reviewer #2: Yes

Reply: Thank you very much for your checking our paper.

12. Reviewer #1, General comments: The paper titled “ACRBP (Sp32) is involved in priming sperm for the acrosome reaction and the binding of sperm to the zona pellucida in a porcine model” has carried out experiments that are well-done methodologically, but it is difficult to understand the fundamental reasons that has led the authors to carry out the experiments. The justification for each experiment is not always present, despite being directly related to sperm capacitation. This study suggests the function of ACRBP in the fertilization process. 

Reply: We previously confirmed the tyrosine phosphorylation was involved on the p32 in the pig sperm by the western blots methods, meaning that the data is the lack of novelty. Therefore, the data of western blots was not indicated and the capacitation rate by CTC stain was only shown in the paper. 

13. Reviewer #1, General comments: The number of replicates is not shown consistently, as it is absent sometimes or given an interval. 

14. Reply: I changed all the figures to tables to clearly show all the value and the number of replicates and statistical values in the tables. I hope that these tables are strong enough to state that the statistical analysis been performed appropriately and rigorously. Please note all the tables.

15. Reviewer #1, General comments: There are many studies on sperm capacitation on porcine species, but these authors only refer to theirs or to other species, especially mice, where studies are carried out with sperm from the epididymis that have not been in contact with the seminal plasma and in many times these results are not comparable. Why have not the authors used more bibliographic references of pigs? In summary, it is a work that could have been better written to make its understanding easier and relying more on authors who have worked on sperm capacitation in pigs.

Reply: I changed some references investigating pig sperm. Please note the reference 17 and 21.

16. Reviewer #1, Specific comments: Line 185 washed once with ml BTS in a 15 ml centrifuge tube – time and centrifugation force?

Reply: It is noted that the time and centrifuge force are 10 minutes and 250 x G, respectively. Please note L189.

17. Reviewer #1, Specific comments: Line 203, should say number of oocytes and replicates used

Reply: I changed all figures to tables. Table 2 indicates number of oocytes and replicates. I hope that this clearly shows that all the tables are strong enough to convince Reviewer #1. Please note all the tables.

18. Reviewer #1, Specific comments: Line 208, should explain what sperm suspension is

Reply: sperm suspension was changed to ``cultured sperm in fertilization medium’’. Please note L212.

19. Line 273, should explain what is the pre-capacitated and pre-incubated boar sperm

Reply: I changed the sentence, “the pre-capacitated and pre-incubated boar sperm” to “washed and uncultured sperm in fertilization medium”. Please note L279.

20. Reviewer #1, Specific comments: Line 242, 250, 275, 288 should say the number of replicates,

Reply: I changed all figures to tables. This mean that all the value and the number of experiment replicates and statistical values are shown in the tables. I hope that these clear table are strong enough to convince Reviewer #1. Please note the tables, L366-L367, L382-L383, L422-L423, L460-L461, L499-500.

21. Reviewer #1, Specific comments: Line 286 The authors wrote “Capacitation and the AR was assessed using PSA-FITC staining” but capacitation was evaluated by CTC. Specify. The IVF results should be more extend and shown in a table, it is important to know how many oocytes were used, how many were penetrated, how many sperm penetrated per oocyte, polyspermy rate, asynchronous penetration... And, of course, the number of replicates included.

Reply: I changed all figures to tables. This mean that all the value and the number of experiment replicates and statistical values are shown in the tables. I hope that these clear tables are strong enough to convince Reviewer #1. Please note the tables, L366-L367, L382-L383, L422-L423, L460-L461, L499-500.

22. Reviewer #2, Referee’s comments: The paper submitted by Kato et al. entitled “ACRBP (Sp32) is involved in priming sperm for the acrosome reaction and the binding of sperm to the zona pellucida in a porcine model” describes the role played by ACRBP on ZP-binding and acrosome reaction in relations with tyrosine phosphorylation and intracellular calcium concentration. The experiments are clearly described, the results sustain the working hypothesis and the paper is globally well written. I only have a few remarks the authors may consider before the paper can be considered for publication. The fact that antibodies linked on the sperm surface inhibit sperm zona-pellucida binding is not a guarantee that the targeted protein is a key player in the fertilization process, as was proven in past for different candidate (such as the Crisp proteins). The most interesting action o ACRBP shown in this paper may thus be the potential interaction with SERCA. In this regards, even though the authors mention in their conclusion that “Future experiments are necessary to assess how ACRBP promotes the activation of the Ca2+ channels on the sperm surface”, it would strengthen the current work to show the co-localisation of these two proteins during the capacitation process. Maybe the authors could perform simple co-localization IF studies using confocal microscopy or proximity ligation assays? This would not impeach a more mechanistic study as proposed in the conclusion (with co-IP and intracellular calcium measurements maybe?).

Reply: Another experiment showing the localization of SERCA and ACRBP were added. I hope that the experiment is strong enough to convince you as more strengthen work. Please note the material and methods in L302-342, the results in L516-L538, and the discussion in L610-622, respectively.

23. Reviewer #2, MAJOR COMMENTS: There are differences in the basal states of acrosome reaction between figures 3 and 4. In the “medium only” condition, the ratio is around 40% in figure 3A whereas it barely reaches 20% in figure 4A. In figure 4A, there is a significant effect when anti-ACRBP antibodies are added, which is not commented in the “results” section. There are also huge differences in the percentages of AR when A23187 is added between these two figures (50% less in figure 4C versus 3C). Maybe the conditions are different, and I did not get a point, unless, could the authors clarify the reasons of these discrepancies?

Reply: Season when the semen was taken, and the experiment were conducted were different between Experiment 3 and 4. Experiment 3 was performed during Winter 2010 and Experiment 4 was conducted in 2011 Spring. Semen quality depend on the temperature where pigs live, and higher temperature negatively affect the semen quality. This mean that sperm viability might decrease. Therefore, the difference of the percentage was explained by the season.

24. Reviewer #2, MAJOR COMMENTS: In figure 2A, the difference between “No antibodies” and “Blocking peptide” is around 50% (40% vs. 20%, respectively) with small SEM as indicated by the error bars. There is no significant difference indicated here, this seems curious… Could the authors verify this point?

Reply: P value is 0.1097 and F value is 2.85 between both data, so there is no difference by one-way ANOVA. 

25. Reviewer #2, MAJOR COMMENTS: Line 430-431, the authors write that “only a few phosphorylated proteins have been formally identified” concerning sperm tyrosine and/or Ser/Thr phosphorylated proteins. Even if the amount of literature is not huge, two papers in mouse spermatozoa report the identification of tyrosine phosphorylated proteins during capacitation (Arcelay et al., 2008 and Chung et al., 2014). Other data are available in different conditions and in different organisms, including Human. If this point needs to be discussed, some more recent and accurate references should be added and commented.

Reply: I agreed with the suggestions. I added several sentences with the references. Please note L556-L563. 

26. Reviewer #2, MINOR COMMENTS: Line 282 the authors indicate the use of 5mM A23187, is that correct? This concentration seems very high and A23187 is more usually used in the micromolar range.

Reply: It was mistakenly noted that the A23187 concentration was 5mM. The concentration in the paper have been changed to 10 uM. Please note L 291.

27. Reviewer #2, MINOR COMMENTS: Lines 394-395 the authors write that “the additions of 0.1% DMSO in the fertilization medium promoted the parameters of the AR (Fig 4 B, P>0.05)”. Is this a mistake? Nothing seems different from the control situation in this figure.

Reply: It is a mistake. I changed ``The additions of 0.1% DMSO in the fertilization medium promoted the parameters of the AR (Fig 4 B, P>0.05) when the group of DMSO was compared with the group of medium only`` to`` The additions of 0.1% DMSO in the fertilization medium did not promote the parameters of the AR (Fig 4 B, P>0.05) when the group of DMSO was compared with the group of medium only ``. Please note L485-L486.

28. In-house officer 1: Please upload a copy of Figure 4 which you refer to in your text. Or if the figure is no longer to be included as part of the submission please remove all reference to it within the text.

Reply: We changed all the figures of the graph to the tables, therefore, the notation of the figures in the sentence to the tables. Please note L361, L363, L404, L406, L408, L410, L445, L449, L451, L481, L483, L485, L486, L489, L554, L589, L603, L605, L633, and L636.

29. In-house officer 2: Thank you for updating your data availability statement. You note that your data are available within the Supporting Information files, but no such files have been included with your submission. At this time we ask that you please upload your minimal data set as a Supporting Information file, or to a public repository such as Figshare or Dryad. Please also ensure that when you upload your file you include separate captions for your supplementary files at the end of your manuscript. As soon as you confirm the location of the data underlying your findings, we will be able to proceed with the review of your submission.

Reply: The supporting Information file of minimal data sets of our findings were uploaded. Please note them in the end of the manuscripts.

30. Additional changes by authors: We changed all the figures of the graph to the tables, therefore, the notation of the figures in the sentence to the tables. Please note L361, L363, L404, L406, L408, L410, L445, L449, L451, L481, L483, L485, L486, L489, L554, L589, L603, L605, L633, and L636.

---

## [Decision Letter · Decision Letter 1]

20 Nov 2020

PONE-D-20-10428R1

ACRBP (Sp32) is involved in priming sperm for the acrosome reaction and the binding of sperm to the zona pellucida in a porcine model

PLOS ONE

Dear Dr. Kato,

Thank you for submitting your manuscript to PLOS ONE. After careful consideration, we feel that it has merit but does not fully meet PLOS ONE’s publication criteria as it currently stands. Therefore, we invite you to submit a revised version of the manuscript that addresses the points raised during the review process (see below).

We look forward to receiving your revised manuscript.

Kind regards,

Joël R Drevet, Ph.D.

Academic Editor

PLOS ONE

Additional Editor Comments (if provided):

Dear authors,

Your manuscript was returned to the two original reviewers who saw the first draft. Although one of the reviewers appears to be satisfied, the second reviewer is not and I must say that I agree with most of his conclusions/comments. Therefore, I stand by my original decision and return the manuscript to you with a request for "major revision". Please note that this is the last offer to review your manuscript. If it is not sufficiently well done, the following decision will be a rejection.

Reviewers' comments:

Reviewer's Responses to Questions

**Comments to the Author**

1. If the authors have adequately addressed your comments raised in a previous round of review and you feel that this manuscript is now acceptable for publication, you may indicate that here to bypass the “Comments to the Author” section, enter your conflict of interest statement in the “Confidential to Editor” section, and submit your "Accept" recommendation.

Reviewer #1: All comments have been addressed

Reviewer #2: (No Response)

2. Is the manuscript technically sound, and do the data support the conclusions?

Reviewer #1: Yes

Reviewer #2: (No Response)

3. Has the statistical analysis been performed appropriately and rigorously? 

Reviewer #1: Yes

Reviewer #2: (No Response)

4. Have the authors made all data underlying the findings in their manuscript fully available?

Reviewer #1: Yes

Reviewer #2: (No Response)

5. Is the manuscript presented in an intelligible fashion and written in standard English?

Reviewer #1: Yes

Reviewer #2: (No Response)

6. Review Comments to the Author

Reviewer #1: (No Response)

Reviewer #2: Referee’s comments:

The paper is a revised version (PONE-D-20-10428R1) of a work originally submitted by Kato et al. and entitled “ACRBP (Sp32) is involved in priming sperm for the acrosome reaction and the binding of sperm to the zona pellucida in a porcine model”.

The authors tentatively replied to the comments made in the first round of reviewing. After reading the new version of the manuscript and the answers of the authors, I am still puzzled by a certain number of things reported below.

- One first point in my major comments was that the fact that antibodies linked on the sperm surface inhibit sperm zona-pellucida binding is not a guarantee that the targeted protein is a key player in the fertilization process, as was proven in the past for different candidate (such as the Crisps proteins). This point has not been discussed by the authors. Are there studies with ACRBP gene invalidation corroborating the putative ZP-binding function of ACRBP?

- Another comment was that “Maybe the authors could perform simple co-localization IF studies using confocal microscopy or proximity ligation assays?” in order to show the co-localisation of the two studied proteins during the capacitation process. The authors performed IF staining of each protein in two independent experiments. The results only show that each protein is present in the acrosome part. Co-localisation studies usually involve co-staining if you use IF, and confocal microscopy with image analysis to prove that the two signals are indeed co-localized. Proximity ligation assay is another possible option, maybe more difficult to perform. In my opinion the authors did not answer that point in a satisfactory way. If the authors believe that the co-localisation is not a crucial point to sustain the function of these proteins, they need to argue in that sense.

Furthermore, the investigation of the co-localisation has not been considered during the capacitation process, as was mentioned in my first report.

- In their answer concerning the discrepancies in the data relating the percentages of acrosome reacted sperm (see Tables 4 and 5 of the revised manuscript), the authors mention that the season is the cause. Nothing has been added in the text. Maybe this point is an evidence for researchers specialized in pig semen, but this is not the case for others although they may work in the reproductive field.

- I suggested two references concerning the identity of identified tyrosine-phosphorylated proteins during capacitation. The authors used other ones, which is not the problem. My concern is that they still mention “only a few phosphorylated proteins, which are hexokinase in the sperm head and the a-kinase-anchoring protein in the flagellum, have been formally identified [43]”. If they mean in porcine, it needs to be clearly stated, because in the two references I have proposed, other proteins were identified (45 proteins in Chung et al., 2014 using phopsphoproteomics analysis). Maybe the authors consider that these proteins were not “formally identified” but they have to argue that point.

- There are very numerous spelling or grammar errors in the added portions of the text, and even a reference in a place where it should not be (lines 324-327). The authors should be more careful when submitting a paper for review. As an indication, I noticed errors in lines 282, 284, 293, 295, 304,307,309, 320, 321, 324-327, 332.334, 361, 363, 400, legend of table 3, 440, 445 (discrepancy between the text and the table legend), 520, 536-537, 563, 564, 573, 575, 668, and probably others that I missed.

Conclusion: Considering the above-mentioned remarks, I cannot change the “major revisions” status that was proposed in the first review.

7. PLOS authors have the option to publish the peer review history of their article (what does this mean?). If published, this will include your full peer review and any attached files.

Reviewer #1: No

Reviewer #2: No

---

## [Author Response · Author response to Decision Letter 1]

11 Apr 2021

Re: PLOS ONE Decision: Revision required [PONE-D-20-10428] [EMID:df198b69de0b9df3]

Title: ACRBP (Sp32) is involved in priming sperm for the acrosome reaction and the binding of sperm to the zona pellucida in a porcine model

Authors: Yoku Kato, Satheesh Kumar, Christian Lessard, Janice L Bailey

Dear Professor Drevet,

Thank you for sending your reviews. We appreciate the helpful suggestions offered by the editor and reviewer as their comments were valuable for improving this manuscript. We have modified our manuscript for PLOS ONE in accordance with comments from editor and reviewers and hereby resubmit the revised manuscript for consideration. In addition, one supporting information file of the additional data was added. These changes have addressed all of critiques of the reviewers. All revisions are listed as followed. 

As the reviewer mentioned the grammatical errors, the attached paper has been carefully reviewed by an experienced editor whose first language is English and who is authorized as an official marker of International English Language Testing System and specialized in the editing of papers written by scientists whose native language is not English. 

We hope that the revised version of our manuscript is now suitable for publication in PLOS ONE and we look forward to hearing from you at your earliest convenience.

Yours sincerely, 

Yoku Kato, Ph.D

Centre de recherche en reproduction, 

développement et santé intergénérationnelle, 

Département des sciences animales,　

Université Laval, Ste-Foy, Québec, Canada

Tel: +81-90-4169-3285

E-mail; hiroisora2001@yahoo.co.jp

Janice L. Bailey, Ph.D 

Fonds de recherche du Québec

140, Grande Allée Est, bureau 450

Québec (Québec) G1R 5M8

Tel: +1-418 643-3230

E-mail: janice.bailey@frq.gouv.qc.ca

[Comments and Questions]

1. Reviewer #2, MAJOR COMMENTS: One first point in my major comments was that the fact that antibodies linked on the sperm surface inhibit sperm zona-pellucida binding is not a guarantee that the targeted protein is a key player in the fertilization process, as was proven in the past for different candidate (such as the Crisps proteins). This point has not been discussed by the authors. Are there studies with ACRBP gene invalidation corroborating the putative ZP-binding function of ACRBP? 

Reply: Revised Introduction section. Please note L108-112: 

“In vitro fertilization (IVF) assays using cumulus-intact oocytes indicated a clear reduction in the capacity of sperm produced by mice lacking the ACRBP gene to fertilize the oocytes. The fertilization rate in the deficient sperm was less than 10% of that of normal sperm. The abilities of the deficient sperm to bind to the zona pellucida and to fuse with the oolemma were significantly decreased [34].”

2. Reviewer #2, MAJOR COMMENTS: In the first review, the reviewer mentioned “the fact that antibodies linked on the sperm surface inhibit sperm zona-pellucida binding is not a guarantee that the targeted protein is a key player in the fertilization process, as was proven in past for different candidate (such as the Crisp proteins). The most interesting action o ACRBP shown in this paper may thus be the potential interaction with SERCA. In this regard, even though the authors mention in their conclusion that “Future experiments are necessary to assess how ACRBP promotes the activation of the Ca2+ channels on the sperm surface”, it would strengthen the current work to show the co-localisation of these two proteins during the capacitation process. Maybe the authors could perform simple co-localization IF studies using confocal microscopy or proximity ligation assays? This would not impeach a more mechanistic study as proposed in the conclusion (with co-IP and intracellular calcium measurements maybe?). “

In the second review, the same reviewer’s comments was that “Maybe the authors could perform simple co-localization IF studies using confocal microscopy or proximity ligation assays?” in order to show the co-localisation of the two studied proteins during the capacitation process. The authors performed IF staining of each protein in two independent experiments. The results only show that each protein is present in the acrosome part. Co-localisation studies usually involve co-staining if you use IF, and confocal microscopy with image analysis to prove that the two signals are indeed co-localized. Proximity ligation assay is another possible option, maybe more difficult to perform. In my opinion the authors did not answer that point in a satisfactory way. If the authors believe that the co-localisation is not a crucial point to sustain the function of these proteins, they need to argue in that sense. Furthermore, the investigation of the co-localisation has not been considered during the capacitation process, as was mentioned in my first report”.

Reply: Thank you very much for these suggestions. Although our ability to conduct the exact experiments suggested by Reviewer #2 is impeded by the pandemic, we have I added another experiment to show the interaction between ACRBP and SERCA. Please note L696-L729 and Figures 3 and 4.

3. Reviewer #2, MAJOR COMMENTS: In their answer concerning the discrepancies in the data relating the percentages of acrosome reacted sperm (see Tables 4 and 5 of the revised manuscript), the authors mention that the season is the cause. Nothing has been added in the text. Maybe this point is an evidence for researchers specialized in pig semen, but this is not the case for others although they may work in the reproductive field.

Reply: I added the sentence in the discussion section. Please note L510-515 in the Results section.

4. Reviewer #2, MAJOR COMMENTS: I suggested two references concerning the identity of identified tyrosine-phosphorylated proteins during capacitation. The authors used other ones, which is not the problem. My concern is that they still mention “only a few phosphorylated proteins, which are hexokinase in the sperm head and the a-kinase-anchoring protein in the flagellum, have been formally identified [43]”. If they mean in porcine, it needs to be clearly stated, because in the two references I have proposed, other proteins were identified (45 proteins in Chung et al., 2014 using phopsphoproteomics analysis). Maybe the authors consider that these proteins were not “formally identified” but they have to argue that point.

Reply: Thank you. The two suggested references are included. Please note L606-609 in the Discussion.

5. Reviewer #2, MAJOR COMMENTS: There are very numerous spelling or grammar errors in the added portions of the text, and even a reference in a place where it should not be (lines 324-327). The authors should be more careful when submitting a paper for review. As an indication, I noticed errors in lines 282, 284, 293, 295, 304,307,309, 320, 321, 324-327, 332.334, 361, 363, 400, legend of table 3, 440, 445 (discrepancy between the text and the table legend), 520, 536-537, 563, 564, 573, 575, 668, and probably others that I missed. 

Reply: As the reviewer mentioned the spelling and grammatical errors, the attached paper has been carefully reviewed by an experienced editor whose first language is English and who is authorized as an official marker of International English Language Testing System and specialized in the editing of papers written by scientists whose native language is not English. Before we submitted the last revision, the editor had corrected the grammatical errors and spelling thoroughly. Please note the lines 27, 28, 29, 32, 34, 37, 69, 74, 75, 77, 79, 80, 84, 88, 98, 99, 101, 102, 104, 114, 115, 120, 121, 151, 152, 153, 156, 157, 166, 172, 174, 175, 179, 183, 184, 185, 186, 193, 198, 199, 200, 201, 202, 206, 207, 213, 217, 218, 227,228, 235, 241, 253, 257, 258, 264, 267, 274, 276, 279, 281, 283, 284, 285, 286, 293, 295, 296, 297, 298, 299, 301, 306, 314, 315, 326, 333, 337, 342, 343, 344, 345, 347, 352, 356, 358, 360, 363, 364, 366, 367, 369, 370, 375, 376, 385, 388, 391, 392, 393, 424, 425, 426, 427, 428, 429, 430, 431, 434, 435, 458, 467(discrepancy between the text and the table legend), 468, 469, 471, 471, 472, 473, 504, 505, 507, 542, 543, 549, 550, 556, 559, 564, 565, 581, 582, 584, 585, 586, 592, 594, 596, 598, 599, 600, 602, 606, 611, 614, 619, 620, 623, 624, 625, 626, 627, 636, 637, 638, 639, 640, 641, 642, 643, 644, 646, 648, 649, 650, 651, 652, 653, 654, 655, 656, 657, 658, 659, 660, 661, 662, 663, 664, 665, 666, 667, 668, 669, 670, 671, 672, 673, 674, 675, 678, 679, 680, 681, 682, 683, 684, 685, 686, 688, 689, 690, 692, 693, 694, 694, 695, 731, 733, 746, 748, Table 1, Table 3, and Table 4. In addition, I realized that the reference on the line 324 and 327 in the previous version of paper is incorrect. The place was corrected and please see the line 326.

---

## [Decision Letter · Decision Letter 2]

26 Apr 2021

PONE-D-20-10428R2

ACRBP (Sp32) is involved in priming sperm for the acrosome reaction and the binding of sperm to the zona pellucida in a porcine model

PLOS ONE

Dear Dr. Yoku Kato,

Thank you for submitting your manuscript to PLOS ONE. After careful consideration, we feel that it has merit but does not fully meet PLOS ONE’s publication criteria as it currently stands. Therefore, we invite you to submit a revised version of the manuscript that addresses the points raised during the review process. See specific points below.

Please submit your revised manuscript ASAP. If you will need more time than this to complete your revisions, please reply to this message or contact the journal office at plosone@plos.org. Please include the following items when submitting your revised manuscript:

We look forward to receiving your revised manuscript.

Kind regards,

Joël R Drevet, Ph.D.

Academic Editor

PLOS ONE

Journal Requirements:

Additional Editor Comments (if provided):

Please answer the points brought forward by the reviewer.

Reviewers' comments:

Reviewer's Responses to Questions

**Comments to the Author**

1. If the authors have adequately addressed your comments raised in a previous round of review and you feel that this manuscript is now acceptable for publication, you may indicate that here to bypass the “Comments to the Author” section, enter your conflict of interest statement in the “Confidential to Editor” section, and submit your "Accept" recommendation.

Reviewer #2: (No Response)

2. Is the manuscript technically sound, and do the data support the conclusions?

Reviewer #2: Yes

3. Has the statistical analysis been performed appropriately and rigorously? 

Reviewer #2: I Don't Know

4. Have the authors made all data underlying the findings in their manuscript fully available?

Reviewer #2: Yes

5. Is the manuscript presented in an intelligible fashion and written in standard English?

Reviewer #2: Yes

6. Review Comments to the Author

Reviewer #2: Referee’s comments:

The paper is a second revised version (PONE-D-20-10428R2) of a work originally submitted by Kato et al. and entitled “ACRBP (Sp32) is involved in priming sperm for the acrosome reaction and the binding of sperm to the zona pellucida in a porcine model”.

The authors replied to the comments made in the second round of reviewing. After reading the new version of the manuscript and the answers of the authors, I only have minor comments:

- The authors wrote lines 540-541: “Indirect immunofluorescence revealed the presence of SERCA 2 at the acrosome (Fig 1, panel A). Then they wrote lines 543-544: Indirect immunofluorescence showed the presence of ACRBP at the acrosome (Fig 1, panel A).

What exactly is presented in Fig 1? It is indicated that it is SERCA in the figure legend.

- The authors wrote lines 600-602: “Our initial experiment demonstrates that anti-phosphotyrosine and anti-ACRBP antibodies significantly reduced sperm-ZP penetration (Table 1 A and C, P<0.05) and fertilization (Figure 1 B and D, P<0.05)”. It seems that there is no Table 1 "A or C" and no Figure1"D" reporting fertilization data.

- Line 635: “Anti-ACRBP antibodies obviously impeded capacitation (Figure 2, P<0.05).” Here also please correct the figure number mentioned.

- Line 731 please correct “is vital for the biding of sperm to the ZP”.

Conclusion: Considering the above-mentioned remarks, I suggest that the paper is acceptable for publication after correcting the mentioned errors.

7. PLOS authors have the option to publish the peer review history of their article (what does this mean?). If published, this will include your full peer review and any attached files.

Reviewer #2: No

---

## [Author Response · Author response to Decision Letter 2]

30 Apr 2021

Rebuttal letter for or PONE-D-20-10428R2

Title: ACRBP (Sp32) is involved in priming sperm for the acrosome reaction and the binding of sperm to the zona pellucida in a porcine model

Authors: Yoku Kato, Satheesh Kumar, Christian Lessard, Janice L Bailey

We would like to thank the editor and the two reviewers for their comments on our manuscript. Our response to each point raised by the academic editor and reviewers are described in the below. We hope that we fully addressed all the concerns, and that the manuscript will be now suitable for publication in PLOS ONE. 

Yours sincerely, 

On behalf of all authors,

Yoku Kato, Ph.D

Centre de recherche en reproduction, 

développement et santé intergénérationnelle, 

Département des sciences animales,　

Université Laval, Ste-Foy, Québec, Canada

Tel: +81-90-4169-3285

E-mail: hiroisora2001@yahoo.co.jp

1. Journal Requirements: 

Reply: I thoroughly checked that there are no retracted references using PubMed. I am very grateful if you tell me about any retracted reference which I missed. 

Reference 82 has been changed reference 80. This is because the reference 81 and 46 are the same reference. Please note Line 1050. 

Reference 60 and 64 are the same reference. The reference 64 was removed and other references were advanced. Please see Line 992 to Line 1052.

Reference 6 and 49 are the same reference. The reference 6 was changed to more suitable reference indicating that increased protein phosphorylation is associated with capacitation, hyperactivated motility, sperm-ZP binding, the AR, and sperm-oocyte binding and fusion. Please note Line 775.

2. Review Comments to the Author 

Reply: These studies were conducted with the approval of the institutional committees for research integrity, notably the Comité de protection des animaux de l'Université Laval and the Comité de gestion des produits chimiques de l’Université Laval. Please note Line 127-129. Those statements were added in the manuscript. Furthermore, we claim that this publication is not dual publication. Previously, we have presented a part of this study in the 2011 Annual Meeting in Society for the Study of reproduction as followed. 

Janice L. Bailey, Yoku Kato, Christian Lessard. Tyrosine Phosphorylation of Sperm Proteins: What Does It Mean? Biology of Reproduction, Volume 85, Issue Suppl_1, 1 July 2011, Page 103.

Reviewer #2: Referee’s comments: 

- The authors wrote lines 540-541: “Indirect immunofluorescence revealed the presence of SERCA 2 at the acrosome (Fig 1, panel A). Then they wrote lines 543-544: Indirect immunofluorescence showed the presence of ACRBP at the acrosome (Fig 1, panel A). What exactly is presented in Fig 1? It is indicated that it is SERCA in the figure legend. 

Reply: Thank you very much for your comments. The line 544 was corrected as Fig 2. “Fig 1, panel A” changed correctly to “Fig 2, panel A”. Please note the line 548. 

- The authors wrote lines 600-602: “Our initial experiment demonstrates that anti-phosphotyrosine and anti-ACRBP antibodies significantly reduced sperm-ZP penetration (Table 1 A and C, P<0.05) and fertilization (Figure 1 B and D, P<0.05)”. It seems that there is no Table 1 "A or C" and no Figure1"D" reporting fertilization data. 

Reply: Thank you very much for your comments. Please note Line 605 and Line 606. “Table 1 A and C” and” Figure 1 B and D” were changed to “Table 1” and “Table 2” individually.

- Line 635: “Anti-ACRBP antibodies obviously impeded capacitation (Figure 2, P<0.05).” Here also please correct the figure number mentioned.

Reply: Thank you very much for your comments. Please note the line 635. “Figure 2.” was changed to “Table 3”

- Line 731 please correct “is vital for the biding of sperm to the ZP”. 

Reply: Thank you very much for your comments. Please note the line 734. “is vital for the biding of sperm to the ZP” was changed to “is vital for sperm-ZP binding”

---

## [Decision Letter · Decision Letter 3]

7 May 2021

ACRBP (Sp32) is involved in priming sperm for the acrosome reaction and the binding of sperm to the zona pellucida in a porcine model

PONE-D-20-10428R3

Dear Dr. Kato,

We’re pleased to inform you that your manuscript has been judged scientifically suitable for publication and will be formally accepted for publication once it meets all outstanding technical requirements.

Kind regards,

Joël R Drevet, Ph.D.

Academic Editor

PLOS ONE

Additional Editor Comments (optional):

Reviewers' comments:

Reviewer's Responses to Questions

**Comments to the Author**

1. If the authors have adequately addressed your comments raised in a previous round of review and you feel that this manuscript is now acceptable for publication, you may indicate that here to bypass the “Comments to the Author” section, enter your conflict of interest statement in the “Confidential to Editor” section, and submit your "Accept" recommendation.

Reviewer #2: All comments have been addressed

2. Is the manuscript technically sound, and do the data support the conclusions?

Reviewer #2: (No Response)

3. Has the statistical analysis been performed appropriately and rigorously? 

Reviewer #2: (No Response)

4. Have the authors made all data underlying the findings in their manuscript fully available?

Reviewer #2: (No Response)

5. Is the manuscript presented in an intelligible fashion and written in standard English?

Reviewer #2: (No Response)

6. Review Comments to the Author

Reviewer #2: (No Response)

7. PLOS authors have the option to publish the peer review history of their article (what does this mean?). If published, this will include your full peer review and any attached files.

Reviewer #2: No

---

## [Editor Report · Acceptance letter]

17 May 2021

PONE-D-20-10428R3 

ACRBP (Sp32) is involved in priming sperm for the acrosome reaction and the binding of sperm to the zona pellucida in a porcine model 

Dear Dr. Kato:

I'm pleased to inform you that your manuscript has been deemed suitable for publication in PLOS ONE. Congratulations! Your manuscript is now with our production department. 

Kind regards, 

on behalf of

Prof. Joël R Drevet 

Academic Editor

PLOS ONE